# Transcriptome Sequencing Analysis of Root in Soybean Responding to Mn Poisoning

**DOI:** 10.3390/ijms241612727

**Published:** 2023-08-12

**Authors:** Ying Liu, Yuhu Pan, Jianyu Li, Jingye Chen, Shaoxia Yang, Min Zhao, Yingbin Xue

**Affiliations:** 1Department of Biotechnology, College of Coastal Agricultural Sciences, Guangdong Ocean University, Zhanjiang 524088, China; 2Department of Agronomy, College of Coastal Agricultural Sciences, Guangdong Ocean University, Zhanjiang 524088, China

**Keywords:** manganese poisoning, root system, soybean, transcription analysis

## Abstract

Manganese (Mn) is among one of the essential trace elements for normal plant development; however, excessive Mn can cause plant growth and development to be hindered. Nevertheless, the regulatory mechanisms of plant root response to Mn poisoning remain unclear. In the present study, results revealed that the root growth was inhibited when exposed to Mn poisoning. Physiological results showed that the antioxidase enzyme activities (peroxidase, superoxide dismutase, ascorbate peroxidase, and catalase) and the proline, malondialdehyde, and soluble sugar contents increased significantly under Mn toxicity stress (100 μM Mn), whereas the soluble protein and four hormones’ (indolebutyric acid, abscisic acid, indoleacetic acid, and gibberellic acid 3) contents decreased significantly. In addition, the Mn, Fe, Na, Al, and Se contents in the roots increased significantly, whereas those of Mg, Zn, and K decreased significantly. Furthermore, RNA sequencing (RNA-seq) analysis was used to test the differentially expressed genes (DEGs) of soybean root under Mn poisoning. The results found 45,274 genes in soybean root and 1430 DEGs under Mn concentrations of 5 (normal) and 100 (toxicity) μM. Among these DEGs, 572 were upregulated and 858 were downregulated, indicating that soybean roots may initiate complex molecular regulatory mechanisms on Mn poisoning stress. The results of quantitative RT-PCR indicated that many DEGs were upregulated or downregulated markedly in the roots, suggesting that the regulation of DEGs may be complex. Therefore, the regulatory mechanism of soybean root on Mn toxicity stress is complicated. Present results lay the foundation for further study on the molecular regulation mechanism of function genes involved in regulating Mn tolerance traits in soybean roots.

## 1. Introduction

Manganese (Mn) is regarded as the vibrant central element of nearly thirty-five different enzymes, such as Mn superoxide dismutase (SOD) and catalase (CAT) [1,2]. These enzymes play key roles in respiratory action, photosynthesis, and protein and hormone synthesis in plants [1]. Although Mn is one of the important trace elements for plant development, excessive Mn may be harmful to plants [3]. In general, plants just need from 20 to 40 mg/kg Mn (dry weight) to maintain normal nutritional requirements, but the content is usually from 30 to 500 mg/kg in most plants [4]. As a result, many plants have higher levels of Mn than required, and excessive Mn may restrain their growth [5]. Mn poisoning has evolved into a serious trouble in the world, resulting in limiting the growth of crops and reducing crop yield in acid soil regions [3,4].

In recent years, plants suffer from Mn poisoning stress more seriously than before due to pollution and acidification of soil. The amount of Mn accumulated in most of the plants has far exceeded their life requirements [6]. Excess accumulation of Mn in plants has a toxic effect on them; for instance, when plants are stressed by Mn toxicity, their growth rate is slowed down, the lateral roots number is reduced, root vitality is decreased, and the activities of various enzymes are inhibited [7,8], thus affecting the formation of multiple hormones [9]. In addition, excessive accumulation of Mn in food crops could threaten human health through the food chain [10], leading to Parkinson’s disease, and influence the regular function of the liver, blood vascular, immunity, and genital systems [11]. Therefore, the damage resulting from Mn poisoning in crops is straightforward and serious, thus restraining their normal growth.

Plants have produced a couple of ways to adapt to Mn poisoning, such as regulating ion absorption, different separation of Mn at the subcellular level, changing the activity of antioxidant enzymes, and promoting generation and excretion of organic acid to transform Mn into its inactive condition [1,2,12]. For example, Mn toxicity increases peroxidase (POD) activity, promotes Mn separation in the extracellular body of pea (*Vigna sinensis*), and causes excessive Mn oxidization [13,14]. Isolation of excessive Mn into the vacuole may play an important role in crops’ response to Mn toxicity [15]. In addition, a number of Mn transport proteins that transfer Mn to the vacuole have been authenticated, such as *Arabidopsis* MTP11, rice MTP8.1, and cucumber MTP8, suggesting that Mn resistance in plants can be adjusted by Mn transport proteins [15,16,17]. Promoted organic acid secretion in roots can promote resistance to excessive Mn via the chelation of excess Mn [18]. In addition, oxalic acid and citric acid secretions in *Lolium perenne* can restrict the absorbing of Mn, thereby enhancing resistance to Mn poisoning stress [19]. However, increased root malic acid secretion had a considerable effect on the tolerance of *Stylosanthes guianensias* to Mn [18]. In addition, roots can promote resistance to Mn poisoning stress by adjusting the absorption of metallic elements, such as Ca, Fe, and Mg [20,21,22].

Soybean (*Glycine max*) is a plant of the genus *Glycine* in the Fabaceae family, and it is one of the most important legumes in the world [23]. As a crop, soybean frequently suffers diverse metal ion stresses throughout its life cycle, not merely limiting the production but threatening people’s health because of accumulating harmful heavy metallic elements [24,25]. Soybeans are highly susceptible to Mn poisoning. When the soluble Mn^2+^ concentration exceeded 50 μM, soybean seedling development was inhibited [11]. Mn stress can affect the absorption and transport of plant ions and lead to the inhibition of root growth, the accumulation of ROS (reactive oxygen species), and the destruction of plant hormone homeostasis in vivo [26,27,28]. Therefore, Mn toxicity is one of the main factors influencing soybean root growth and restricting soybean production. How to improve soybean’s adaptability to Mn poisoning stress has evolved into a main problem that urgently needs to be solved.

For the past few years, RNA sequencing (RNA-seq) has been widely used to explore the mechanism of plants, including the response of *Citrus sinensis*, *Citrus grandis*, *Manihot esculenta*, and *Cucumis melo* to toxicity stress of Al, Cu, Fe, Ca, and other heavy metals [29,30,31,32]. Nevertheless, few studies have been reported on Mn poisoning in soybean, and the molecular regulation mechanism of Mn stress in soybean remains unclear. In the present study, high-throughput deep sequencing was used to conduct whole-genome transcriptome analysis of soybean roots and comparative analysis of Mn toxicity response genes in roots. In addition, the physiological response indices and the contents of various metal ions in roots treated with different concentrations of Mn were studied. The findings revealed further information about regulating specific signaling pathways that participate in adjusting the root resistance to Mn poisoning. They may offer a preliminary foundation for the further study of genes’ functions of resistance to Mn poisoning. Through the above research methods, this study will explore the molecular mechanism of soybean root response to Mn poisoning, further discover some key genes playing regulatory roles in Mn toxicity stress, and lay a foundation for cultivating high-quality soybean varieties with Mn toxicity tolerance traits.

## 2. Results

### 2.1. Effects of Growth of Soybean Root Suffering Mn Poisoning 

Soybean was cultured in a nutrient solution with different exogenous Mn concentrations (5, 50, 100, 150, 200, and 300 µM) for 15 days. The data of root area index, total root length, root mean diameter, total root volume, and root tips number of soybean were determined to probe the effect of Mn poisoning on root phenotype. The results revealed that with the improvement in exogenous Mn concentrations, Mn toxicity stress had a significant inhibitory effect on soybean root growth (Figure 1A–F and Figure 2A–E). Compared with the soybean roots with normal Mn concentration (5 μM) in the environment, those with high Mn concentrations (50, 100, 150, 200, and 300 µM) were significantly inhibited in terms of soybean root growth (Figure 1A–F). The total root length was significantly reduced by 25.5, 40.4, 55.3, 57.3, and 59.7%, respectively (Figure 2A). The root surface area was also significantly reduced by 28.4, 42.6, 59.0, 58.6, and 65.7%, respectively (Figure 2B). On the contrary, the root volume was decreased significantly by 23.9, 39.3, 57.8, 55.4, and 67.7%, respectively (Figure 2C). The root tips number was also decreased markedly by 24.2%, 37.1, 51.2, 53.6, and 59.7%, respectively (Figure 2E). Meanwhile, the root diameter did not change significantly (Figure 2D). As shown in Figure 1, when the Mn concentration of the treated soybean roots exceeded 50 µM, the root growth was inhibited, and when the treatment concentration was 100 µM, the inhibitory effect was further aggravated.

### 2.2. Influences of Various Exogenous Mn Concentrations on Soybean Root Biomass

The indices of dry and fresh weights of soybean roots were determined under different levels of Mn concentration. The results showed that the fresh root weight and dry root weight were decreased with improvement in the concentrations of exogenous Mn (Figure 3). Compared with the 5 µM treatment, the 50, 100, 150, 200, and 300 µM treatments had significantly decreased root fresh weight by 20.9, 40.0, 49.3, 52.7, and 59.4%, respectively (Figure 3A), and the dry root weight by 23.6, 42.4, 48.5, 59.1, and 60.2%, respectively (Figure 3B). As shown in Figure 3, when the Mn concentration of the treated soybean roots exceeded 50 µM, the root biomass was decreased, and when the treatment concentration was 100 µM, the decreased amount of root biomass was pretty obvious.

### 2.3. Influences of Different Concentrations of Exogenous Mn on the Physiological Response Indices of Soybean Roots

The antioxidant enzymes were subjected to the stress response of Mn poisoning. The reaction trend of these four physiological indices in the root system was basically the same as that of Mn poisoning, and each physiological index increased with the increase in exogenous Mn concentration (Figure 4). Compared with the 5 µM treatment, the 50, 100, 150, 200, and 300 µM treatments significantly increased the POD activity of roots by 54.8, 75.4, 132.9, 174.7, and 177.1%, respectively (Figure 4A); the SOD activity by 268.4, 364.9, 470.2, 700.0, and 836.8%, respectively (Figure 4B); and the APX activity by 60.3, 84.5, 119.0, 136.2, and 208.6%, respectively (Figure 4D).

With the increase in Mn concentrations from 5 μM to 100, 150, 200, and 300 μM, the proline (Pro) content in the roots was significantly increased by 61.7, 85.6, 110.5%, and 104.5%, respectively (Figure 4E); and the CAT activity was significantly increased by 200.0, 342.7, 771.4, and 871.4%, respectively (Figure 4C). Compared with the 5 µM treatment, the 50, 100, 150, 200, and 300 µM treatments decreased the soluble protein content by 4.1, 9.1, 16.3, 18.0 and 20.0%, respectively (Figure 4G), but significantly increased the soluble sugar content by 13.3, 35.5, 42.7, 49.3, and 52.7%, respectively (Figure 4H). The malondialdehyde (MDA) content was significantly increased by 14.1, 25.6, and 45.9% with the increase in Mn concentrations from 5 μM to 150, 200, and 300 μM, respectively (Figure 4F).

### 2.4. Effects of Different Concentrations of Exogenous Mn on Ion Accumulation in Root of Soybean 

After the soybean roots were separately cultured in nutrient solution with 5 and 100 µM Mn for 15 days, the content of Mn, Zn, Ca, Fe, Na, K, Se, Mg, Al, and Cu was determined to explore the influence of Mn poisoning on the amassing of ions in the roots of soybean. The experimental results revealed that with the improvement in exogenous Mn concentration, Mn toxicity stress had an effect on the accumulated content of ions in soybean root (Figure 5A–J). Compared with the content of Mn, Fe, Na, Se, and Al in the roots of soybean under normal Mn concentration (5 μM), those under high Mn concentration (100 μM) were significantly increased by 744.1, 97.5, 84.0, 502.3, and 32.2%, respectively (Figure 5A,C,E,H,I). Furthermore, the contents of Ca and Cu were increased by 5.7 and 1.2%, respectively (Figure 5D,J), whereas those of Zn, K, and Mg were decreased significantly by 14.6, 6.4, and 17.1%, respectively (Figure 5B,F,G).

### 2.5. Influences of Various Concentrations of Exogenous Mn on Plant Hormone Homeostasis in Roots of Soybean 

After the roots of soybean were cultured for 15 days in a nutrient solution with different exogenous Mn concentrations (5 and 100 µM), the hormone content indices were determined to verify the effect of Mn poisoning on root hormones. The experimental results indicated that with the improvement in exogenous Mn concentration, Mn toxicity stress had remarkable effects on the content of hormones in soybean roots (Figure 6). Compared with soybean roots under normal Mn concentration (5 μM), soybean root hormone content was significantly decreased under high Mn concentration treatment (100 µM), and the indole butyric acid (IBA), indoleacetic acid (IAA), gibberellic acid 3 (GA_3_), and abscisic acid (ABA) contents were significantly reduced by 18.4, 21.6, 57.5, and 15.9%, respectively (Figure 6A–D).

### 2.6. RNA-Seq Analysis of Soybean Roots Treated with Mn Poisoning

The transcription levels of soybean roots treated with 5 and 100 μM Mn for 15 days were analyzed by genome-wide RNA sequencing to explore the molecular-level response of roots to Mn poisoning. Six libraries were established from the transcriptome sequencing data of soybean roots treated with the two Mn concentrations. These libraries produced about 40.8–51.5 million base reads, from which about 40.8–51.4 million clean base reads were obtained. Among them, the rate of mass value greater than 30 (Q30) basis was 92.9–94.7% (Appendix A). Finally, 45,274 genes were found in soybean roots, among which 1430 differential expression genes (DEGs) were authenticated in the roots of soybean treated with the two Mn concentrations (Figure 7 and Appendix A). Among these DEGs, 572 were upregulation genes and 858 were downregulation genes (Figure 7 and Appendix A).

### 2.7. Functional DEGs Enrichment Analysis 

The gene ontology (GO) concentration of DEGs in soybean roots was analyzed in accordance with biological process (BP), cell composition (CC), and molecular function (MF). For each GO classification, the most remarkable enrichment was surveyed and presented, as shown in Figure 8 and Appendix A. The results indicated that the DEGs in soybean root were mainly concentrated on MF, followed by BP and CC.

The Kyoto Encyclopedia of Genes and Genomes (KEGG) enrichment site map of DEGs was a visual exhibition of KEGG enrichment analysis results (Figure 9 and Appendix A). In this study, the top 20 pathways with marked enrichment were chosen to be exhibited. If less than 20 pathway entries were enriched, all pathway entries were displayed. The KEGG enrichment was evaluated by Rich factor (RF), FDR, and the quantity of genes gathered in this passageway. RF means the percentage between the amount of DEGs situated in the passageway entry and the total quantity of genes situated in the passageway entry among all of the interpreted genes. The higher the RF is, the higher the gathering level. In general, the value scope of FDR is 0–1, and the more it approaches zero, the more remarkable the enrichment is. In soybean roots, enrichment was mainly manifested in the biosynthesis of secondary metabolites (194 genes, containing 44 upregulation and 150 downregulation genes), isoflavone biological synthesis (31 genes, containing 2 upregulation and 29 downregulation genes), and flavonoid biological synthesis. Furthermore, enrichment was mainly manifested in ubiquinone and other terpenoid–quinone biosynthesis (30 genes, including 1 upregulated and 29 downregulated); cytochrome P450 metabolizes allobiotin (36 genes, containing 2 upregulation and 34 downregulation genes); drug metabolism—cytochrome P450 (35 genes, containing 1 upregulated and 34 downregulated genes); drug metabolism—other enzymes (35 genes, containing 2 upregulation and 33 downregulation genes); metabolic pathway (236 genes, containing 60 upregulation and 176 downregulation genes); glutathione metabolism (35 genes, including 2 upregulated and 33 downregulated genes); biosynthesis of phenylpropanes (36 genes, containing 10 upregulation and 26 downregulation genes); and circadian rhythm—plants (35 genes in total, including 1 upregulated and 34 downregulated genes).

### 2.8. Identification of Hormone DEGs

Among all DEGs, 22 genes relevant to the synthesis of hormones were found in soybean roots (Table 1). Nine genes relevant to synthesis of auxin were found, containing *Glyma.08g010400*, *Glyma.06G134000*, *Glyma.02g142600*, *Glyma.03g029600*, *Glyma.05g101300*, *Glyma.09g011200*, *Glyma.19G258800*, *Glyma.17G046000*, and *Glyma.13G361200*. Moreover, two genes related to ABA synthesis (*Glyma.07G268400* and *Glyma.09G014700*) and one gene related to SA synthesis (*Glyma.16G145100*) were identified. All the DEGs related to hormone synthesis were downregulated in soybean roots. In addition, 10 DEGs related to GA_3_ synthesis were found. Among them, four (*Glyma.12G216100*, *Glyma.12G137700*, *Glyma.02G010100*, and *Glyma.13G285400*) were downregulated, whereas the other six DEGs (*Glyma.08G208500*, *Glyma.13G259500*, *Glyma.19G133600*, *Glyma.16G200800*, *Glyma.19G022500*, and *Glyma.09G149200*) were upregulated.

### 2.9. Identification of Antioxidant DEGs

A total of 14 antioxidant-related DEGs were found in soybean roots treated with normal and high Mn concentrations (Table 2). All of the 14 DEGs were associated with POD. Among them, 2 (*Glyma.18G055500* and *Glyma.17G198700*) were upregulated, whereas the other 12 (*Glyma.03G038200*, *Glyma.20G169200*, *Glyma.16G055900*, *Glyma.20G001400*, *Glyma.01G130800*, *Glyma.03G038700*, *Glyma.01G130500*, *Glyma.10G191700*, *Glyma.09G022800*, *Glyma.09G284400*, *Glyma.02G234200*, and *Glyma.14G104400*) were downregulated.

### 2.10. Identification of Transporter DEGs

A total of 27 transport-associated DEGs were identified, including 1 metal tolerance protein (*Glyma.09G122600*), 1 sulfate transporter gene (*Glyma.18G168900*), 5 vacuolar iron transporter genes (*Glyma*.*02G082500*, *Glyma.08G076000*, *Glyma.08G075900*, *Glyma.05G121300,* and *Glyma.11G083300*), 1 boron transporter gene (*Glyma.09G031400*), 1 magnesium transporter gene (*Glyma.05G153000*), 1 calcium ion transporter gene (*Glyma.19G038600*), 2 ammonium ion transporter genes (*Glyma.07G153800* and *Glyma.20G082450*), 1 aluminum-activated malate transporter gene (*Glyma.19G199900*), 1 metal transporter Nramp5 (*Glyma.06G115800*), 1 molybdate ion transporter (*Glyma.06G074100*), 11 calcium-ion-binding protein genes, and 1 phosphate transporter gene (Table 3). These DEG families showed different expressing patterns in the roots of soybean. For example, the DEGs of the metal tolerance protein, sulfate transporter, vacuolated iron transporter, boron transporter, magnesium transporter, aluminum-activated malic acid transporter, metal transporter Nramp5, and molybdate ion transporter showed upregulated expression. By contrast, the DEGs encoding calcium ion transporters, ammonium ion transporters, calcium-ion-binding proteins, and phosphate transporters were downregulated.

### 2.11. Identification of DEGs of Transcription Factors

A total of 94 DEGs of transcription factors were identified in all DEGs of soybean root (Table 4), including 37 *WRKY transcription factors*, 10 *bHLH transcription factors*, 17 *MYB transcription factors*, 16 *ethylene transcription factors*, 2 *GATA transcription factors*, 2 *heat-stress transcription factors*, 1 *ORG transcription factor*, 2 *NAC transcription factors*, 1 *iron-deficiency-inducible transcription factor*, 1 *TGA transcription factor*, 1 *E2FE transcription factor*, 1 *WER transcription factor*, 1 *JUNGBRUNNEN transcription factor*, and 2 *RAX transcription factors*. The DEGs of these transcription factor families showed different expression patterns in soybean roots. For example, the DEGs that identified *WRKY transcription factors* were downregulated. Among the 10 *bHLH transcription factors*, 4 *transcription factor genes* (*Glyma.11G043700*, *Glyma.15G170500*, *Glyma.09G064200*, and *Glyma.01G197900*) were downregulated, whereas the other 6 genes were upregulated. The expression of 17 *MYB transcription factor* DEGs were downregulated except for *Glyma.15G259400*. Furthermore, 3 (*Glyma.02G067600*, *Glyma.11G014200*, and *Glyma.10G119100*) of the 16 *ethylene transcription factor* DEGs were upregulated, whereas the other 13 genes were downregulated. The DEGs of other transcription factors, including one *GATA gene* (*Glyma.11G068700*), one *ORG gene* (*Glyma.19G132500*), *one iron-deficiency induction gene* (*Glyma.12G178500*), one *E2FE gene* (*Glyma.17G093600*), and one *RAX gene* (*Glyma.07G126900*), were upregulated, whereas the other *transcription factor genes* were downregulated.

### 2.12. Analysis of Expression of Genes by Quantitative Reverse Transcription-PCR (qRT-PCR) 

Quantitative reverse transcription-PCR (qRT-PCR) was implemented for 30 DEGs from the 5 µM (normal group) and 100 µM Mn (treatment group) groups, including 14 genes associated with ion transporters, 4 genes associated with hormones, 3 genes associated with antioxidant enzymes, and 9 genes that act as transcription factors, to confirm the results of transcriptome sequencing (Figure 10).

On the basis of the qRT-PCR analysis, among the gene transcripts evaluated, 30 genes were significantly upregulated or downregulated in the root resistance to Mn poisoning, whereas 10 genes were not (Figure 10). Moreover, two genes related to ion transporters were significantly upregulated, including *metal transporter* (*Nramp5*) and *vacuolar iron transporter homolog 4* (*VIT4c*). Meanwhile, six genes were significantly downregulated, including *calcium-binding proteins* (*CML19*, *CML18*, *CAST*, and *KIC*), *ammonium transporter* (*AMT2.1*), and *calcium-transporting ATPase 13* (*CTATP13*, Figure 10). Six genes related to ion transporters were upregulated, but not significantly, in response to Mn poisoning, including *boron transporter 2 isoform D* (*BT2D*), *magnesium transporter* (*NIPA2*), *metal tolerance protein 10* (*MTP10*), *sulfate transporter 2.1* (*SUT2.1*), and *calcium-binding proteins* (*CBE* and *CML23*, Figure 10). Moreover, four hormone-related genes were significantly downregulated in response to Mn poisoning, including *auxin reactive proteins* (*SAUR36* and *SAUR71*), *gibberellin-responsive protein 1* (*GRP1*), and *ABA synthesis-related protein* (*ABA11E*) (Figure 10). *Peroxidase 46 (POD46*) was significantly downregulated, but *POD4* and *POD16A* were not significantly upregulated or downregulated (Figure 10). Six transcription factor genes were significantly downregulated in response to Mn poisoning, including *bHLH transcription factor* (*bHLH25)*, *ethylene reaction transcription factors* (*ERF017a* and *ERF017b*), *WER transcription factors* (*WER-X2*), *MYB transcription factor* (*MYB14*), and *heat stress transcription factor* (*HS6*); and only *FER transcription factor* (*FER*) was significantly upregulated (Figure 10). These results supported the consequences of the RNA-seq (Appendix A).

## 3. Discussion

Mn poisoning harms crops and restricts agricultural development, especially in acidic soils [1,8]. In general, different crops have different endurance capacity levels on Mn toxicity. For example, soybeans are more vulnerable to excess Mn stress than *Arachis hypogaea* and *Stylosanthes guianensis* [8,33]. Excess Mn inhibited the normal growth of *A. hypogaea* and *Cucumis sativus*, resulting in significantly reduced root dry weight [8,34]. The biomass of *A. hypogaea* root decreased seriously, and the development of the roots was obviously hindered when suffering Mn poisoning [8]. In the present study, the root growth of soybeans was restrained (Figure 1 and Figure 2), and the root biomass decreased significantly after suffering from Mn toxicity (Figure 3). Excess Mn accumulation in plants appears to cause serious cell damage [8], eventually affecting the proper growth of soybean root.

Plants respond to metal ion stress via refraining from excessive metal-ion absorption and restricting metal-ion transport to shoots [35,36]. In certain plant species, the roots retain a high quantity of Mn, whereas only a slight amount is transported to the shoots [37,38]. Another method is linked to the toleration of plant tissue, with excessive metallic elements compartmentalized in saccules or complexed with organic substances [39,40]. In the present study, the Mn concentrations of soybean roots increased when suffering Mn poisoning (Figure 5). Such an increase can be attributed to one of the tactics of the root to fight Mn poisoning via active absorption and storing extra Mn in roots to promote a stress response in soybean.

In this work, excessive Mn was found in the water culture, and the findings revealed that the Mn concentration in roots exceeded sevenfold, which may be the source of Mn poisoning in the roots of soybean (Figure 5). Analogous findings were obtained by Chen in 2016, where two times the Mn content in roots greatly affected the biomass and root growth of the soybean roots [33]. Though it is a necessity, Mn is a kind of metallic element that will be harmful to plants when occurring in high quantities [1]. Mn toxicity can develop once the Mn level in plant shoots exceeds the concentration of 150 mg/kg dry biomass, particularly in acid soil [34,41]. Furthermore, high Mn can block auxin production, limit meristematic cell growth in roots [42], and increase the production of oxidization Mn and phenols in the plant apoplast [14]. These phenomena could further impede root development and dramatically diminish growth quantity [43]. In this research, Mn poisoning of soybean roots was induced as a result of the considerable increase in accumulated Mn content (Figure 5).

Excessive Mn supply significantly increased the Al level in plant roots, and the added Al, in turn, prevented the further absorption of Mn, thus leading to alleviated Mn toxicity [44,45]. Furthermore, Al reduced the contents of Mn in wheat (*Triticum aestivum*) [46], cowpea (*Vigna unguiculata*) [47], and rice (*O. sativa*) [48]. Al may reduce Mn toxicity in rice by lowering symplastic transport of roots in terms of Mn absorption and the effective Mn reserved in roots and shoots [48]. Present results showed that when the soybean root was subjected to Mn poisoning, the concentration of Al in the root could also increase significantly (Figure 5), which may be because of the root symplastic Mn and Al uptake. However, when Al accumulated to a certain amount, it could prevent further accumulation of Mn, thus alleviating Mn poisoning in soybean roots.

In addition, increasing Fe accumulation and improving Fe absorption are beneficial to plant adaptation to Mn toxicity stress [1]. For example, Mn-resistant cotton (*Gossypium hirsutum*) varieties have higher Fe content than Mn intolerant cotton varieties [49]. After being poisoned by Mn, the content of Fe in the soybean root obviously changed and maintained a high content (Figure 5). Therefore, the root system of soybean maintains a high Fe content to alleviate the influence of Mn poisoning, and it may be a physiological response mechanism for soybean to adapt to Mn poisoning.

Differing from the change rule of Mn accumulation content in plants, the content of Mg in soybean roots decreased with the increase in concentration of Mn (Figure 5), indicating that the absorption of Mn and Mg may be antagonistic. Results have displayed that Mn poisoning can inhibit root absorption of Mg in *Sorghum bicolor*, *Solanum lycopersicum*, *S. guianensis*, and other vegetation, causing a noticeable reduction in the concentration of Mg in plants [50]. In this research, Mn poisoning observably decreased the concentration of Mg in roots (Figure 5), suggesting that Mn poisoning mainly hindered the accumulation of Mg by the roots of soybean. Reducing the Mg content in roots may have an important effect in alleviating Mn toxicity and maintaining root function normally, and it may be one of the soybean adaptation processes to Mn poisoning. Studies have shown that excessive Mn in plants can affect the absorption of other important metal nutrients [51,52]. Present studies indicated that the exposure of roots to Mn poisoning seriously affected ion accumulation in the roots of soybean. The molecular regulation mechanism involved is very complex, and it may involve many different types of proteins or genes.

As a secondary metabolite from cell metabolic activity in plants, ROS play a positive role in a plant’s ability to tolerate external stress, which depends on the subtle balance between the production of ROS and removal [53]. APX, POD, and SOD are the antioxidant enzymes taking charge of removing plant ROS [54,55]. By increasing the vitality of oxidation resistance enzymes to decrease the production of intracellular ROS, plants can enhance their tolerance to adversity stress [54,55]. For example, *Broussonetia papyrifera* can effectively alleviate oxidative stress and reduce ROS accumulation by increasing the activity of antioxidases, such as POD, CAT, and SOD [43]. Under the experimental conditions of this study, the enzymatic activities of POD, SOD, CAT, and APX in the soybean root were enhanced significantly when responding to Mn poisoning (Figure 4). Hence, the defense system of oxidation resistance in soybean roots was promptly activated, and the vitalities of various types of antioxidant enzymes in the roots were significantly different. SOD, POD, CAT, and APX may be mainly responsible for scavenging ROS in soybean roots, thus reducing the damage of ROS to soybean roots.

In addition, MDA content is an important reference index to fully reflect the lipid peroxidation level of plant cell membranes [56]. In the present study, the level of peroxidation of the membrane lipid in the roots of soybean suffering Mn poisoning markedly exceeded the normal group (Figure 4). Mn poisoning may destroy the ROS metabolite homeostasis between cells, resulting in an evident increase in the MDA content of roots. Pro plays an important role in regulating osmotic pressure, keeping cell strength, and retaining cytoplasmic stabilization, thereby contributing to holding the cell stability [57]. Pro can also act as one of the ROS removers, working in concert with antioxidase to decrease plant ROS [58]. In the present study, the Pro content in the soybean root exposed to Mn toxicity significantly increased, indicating that soybean roots require more Pro to maintain normal osmotic pressure and remove excessive ROS. Soluble protein can participate in osmoregulation as an osmoregulatory factor, reflecting the degree of plant organ damage [59]. The biosynthesis of soluble protein in plants is influenced by abiotic stress responses [60]. In the present study, the soluble protein content of soybean roots decreased markedly when suffering Mn poisoning, and such a decrease may be a response of soluble protein to Mn poisoning. Moreover, with a view that the degradation of soluble proteins could produce plenty of free amino acids, Pro is a kind of amino acid that increases rapidly in many plants, and its increase may be concerned with the decomposition of soluble proteins [61]. In addition, the contents of soluble sugar showed a significantly increasing trend under Mn poisoning stress in the present study. This finding may be caused by the increase in soluble sugar to regulate cell osmotic pressure and organic small molecule solutes to reduce intracellular water potential to achieve the purpose of absorbing water from the surrounding cells [43]. Furthermore, soluble sugar can stabilize the colloidal properties and protect plant cells from harm [62].

Phytohormones have the important function of controlling the growth of plants and reacting to various abiotic and biotic stressors [9]. Furthermore, growth hormones have an important effect in controlling the growth of roots and reacting to environmental changes; genes from the families of *AUX1/LAX* (*AUXIN1/LIKE AUX1*) and *PIN* (*PINFORMED*) participate in regulating polarity transport in growth hormones and optimize hormone distribution in root tips, which is vital for adjusting root development [42,63]. When *Arabidopsis* seedlings are treated with Cd, the expression of *IAA17* and *PIN1/3/7* is affected distinctly, resulting in the transportation of plant hormones being inhibited in roots and decreasing the hormone contents in the root tip [64]. As a result, the primary root elongation in *Arabidopsis* was inhibited. Aux/IAA proteins are growth hormone signaling regulators, and their stability indicates the plant’s response strategy to environmental changes [65,66]. Superabundant Mn inhibited the development of primary roots in *A. thaliana* [67,68], possibly due to decreased biological synthesis of IAA, and reduced the expression of hormone transporter genes (*PIN4* and *PIN7*) when suffering Mn poisoning; ultimately, this affected the concentration of hormones in the root tips and root development [42]. In this study, the growth hormone levels in soybean roots were less with Mn poisoning than at a normal Mn level (Figure 6), limiting the root development of soybean (Figure 1, Figure 2 and Figure 3).

Gibberellic acid 3 (GA_3_) is a kind of tetracyclic diterpenoid which contributes to plant growth, development, and response to abiotic stressors [69]. GA_3_ increases germination of seeds, elongation of the stem, promotes flowering, expansion of leaf, and the development of fruit [70]. Many genes are involved in controlling GA_3_ production and degradation, including the three main oxidizing enzyme genes in the GA family, namely, *GA20-oxidase*, *GA3-oxidase*, and *GA2-oxidase*, all of which contribute to GA_3_ catabolism and biosynthesis [71]. Root elongation is dependent on the action of GA_3_, which facilitates root cell division and elongation [72]. Furthermore, the GA_3_ content of *Arabidopsis* can influence drought tolerance [73]. GA_3_ can improve *Arabidopsis* salt tolerance by modulating SA levels [74]. Moreover, GA_3_ may reduce Cd poisoning via decreasing the expression of *IRT1* (Cd absorption-related gene) in *A. thaliana* [75]. When the roots of soybean suffered from Mn poisoning, the concentration of GA_3_ in the roots decreased (Figure 6), restraining root elongation (Figure 1–3). This finding could be attributed to Mn poisoning affecting the expression of genes in connection with GA_3_ production, whereafter influencing the concentration of GA_3_ and the growth of root cells.

Abscisic acid (ABA) is a kind of critical hormone which causes plants to tolerate metal poisoning [76]. ABA acts primarily in the meristem and elongation area of roots, and at a certain concentration, ABA stimulates growth of primary roots while inhibiting lateral root development, hence changing the root system conformation [77,78]. Under abiotic stress, plants may adjust the concentration of ABA by modulating the expression of numerous genes involved in the production of ABA [76]. In response to environmental stress, genes such as *abscisic aldehyde oxidase*, *alcohol dehydrogenase* (*ABA2*), and *9-cis-epoxycarotenoid dioxygenase* affect ABA production [76,79,80]. Heavy metal element stressors from Zn, Ni, Al, and Cd can reduce plant ABA concentration [80]. Furthermore, ABA has an important role in dealing with various abiotic stimuli, including metal-ion toxicity, high- or low-temperature stress, drought stress, and saline–alkali stress [80]. In response to the harmful effects of excessive metal ions, ABA may function as a trigger to promote the amassing of Pro and oxidation resistance enzymes [81]. In this research, as the roots of soybean were suffering from Mn poisoning, the genetic expression in connection with ABA biosynthesis in the system of roots was altered (Table 1). Such alteration affected the ABA content (Figure 6) and root development (Figure 1, Figure 2 and Figure 3), most likely on account of the cell development of the meristem and elongation region in the roots of soybean influenced by the concentration of ABA. As a result, the formation of the root structure was altered.

Although RNA-seq has been used to identify the response of some plants to heavy-metal ion stress, only the responses of grape (*Vitis vinifera*) roots and peanut roots to Mn toxicity stress have been reported [8,82,83,84]. A total of 2629 and 3278 DEGs were found in the roots of Combier and Jinshou cultivars, respectively, indicating that these two cultivates may have a different tolerance to Mn toxicity [85]. Transcriptomic profiling studies revealed 731 DEGs in peanut roots suffering Mn poisoning, showing the response of peanut roots to Mn poisoning [8]. At present, genome-wide studies mainly focus on the response of soybean leaves to Mn poisoning [2,86]. However, no reports are available on the response of roots to Mn poisoning in soybean. In the present work, a genome-wide study of the DEGs responding to Mn poisoning in the roots of soybean was conducted using RNA-seq technology. A total number of 1430 DEGs were found in the soybean roots. Among them, 572 and 858 genes were upregulated and downregulated, respectively (Figure 7). Therefore, soybean roots may have unique molecular regulatory pathways to cope with Mn toxicity stress.

Excessive Mn transfer in many plants is a response mechanism to Mn toxicity stress, and this process is mainly regulated by metal-ion transporters [22,87]. For example, the expression of *Prunus persica PpNramp5* was considerably enhanced under Mn poisoning stress, and the heterologous expression of *PpNramp5* in the *Saccharomyces cerevisiae* verified the function of transporting Mn [88]. Wheat (*Triticum aestivum*) vacuolar iron transport protein TaVIT2 is involved in the transportation of Mn; and the study shows that by promoting iron transport in the endosperm vacuole, this indispensable trace element accumulates in the tissue [89]. In the present work, metal-ion transporter genes *GmNramp5* (*Glyma.06G115800*) and *GmVIT4c* (*Glyma.08G076000*) were significantly upregulated in the soybean roots suffering Mn poisoning (Table 3), indicating that they were responsive to Mn toxicity. Therefore, altering excessive Mn transport in soybean roots by altering the transcription of metal-ion transporters may be crucial to Mn tolerance in soybean.

Subcellular localization of Mn has an important effect in plants’ tolerance to Mn toxicity [34]. Studies have suggested that *A. thaliana* AtECA3 and AtECA1 are located in the Golgi apparatus and ER (endoplasmic reticulum), respectively, belonging to Ca-transporting ATPase and directly regulating the transport of excessive Mn to these two locations [90,91]. Furthermore, in a Mn-toxic environment, *AtECA1* or *AtECA3* mutation can hinder development of roots and chlorophyll synthesis of *A. thaliana* [90,91]. In the current study, the expression of calcium transport ATPase (*Glyma.19G038600*) in the roots of soybean was downregulated to adapt to Mn poisoning (Table 3), suggesting that the calcium transport ATPase gene in soybean root may be involved in the detoxification of Mn. It may also be located in the endoplasmic reticulum or Golgi apparatus and be involved in adjusting the division and regionalization of excess Mn in these two locations.

In addition, the family of MTP is a kind of crucial Mn transporter protein that controls the absorption and transport of Mn in plants [92,93]. Research shows that *Camellia sinensis CsMTP8.2* acts as a Mn specific transporter protein, promoting excess Mn^2+^ outflow from vegetable cells [93]. Moreover, *O. sativa* OsMTP11 is involved in Mn activization in cytoplasmic and vacuolar membranes, and it may play vital effects in transporting Mn and other metallic elements [92]. In the present study, *GmMTP* (*MTP10*) was found in the roots in response to Mn poisoning, which caused its expression (*Glyma.09G122600*) to be upregulated (Table 3). The finding suggested that *GmMTP10* may participate in the adaptability of soybeans to Mn poisoning by regulating the accumulation and redistribution of Mn, possibly providing molecular-level evidence for the apparent Mn symptoms found in soybean roots.

Maintaining the dynamic balance of ions in plants, possibly by regulating the absorption and transport of metal elements, such as Fe and Mg, is a critical method to deal with Mn poisoning stress [2,8]. This study indicates that 100 µM of Mn treatment increased the expression of the magnesium transporter gene (*Glyma.05G153000*) in soybean roots (Table 3). Such an increase may promote a large amount of Mg in the roots to be transported from the roots to other tissue sites of the plant, thus resulting in a substantial decrease in Mg content in the roots. This finding indicated that Mn poisoning stress may affect the gene expression of the Mg transporter. Therefore, the accumulation of Mg in soybean roots was affected. In addition, the elevation of Fe can ameliorate Mn toxicity, to some extent, in barley (*Hordeum vulgar*) [94]. Fe also helps improve the tolerance of *Triticum aestivum* to excess Mn stress [8,95]. In this research, excess Mn stress improved the content of Fe in soybean roots. The increased Fe may have a positive effect on the adaptation of soybean roots to excess Mn stress.

Because excess Mn may bring about oxidative responses of stress, adjusting the antioxidant enzyme activity is generally considered as a basic method of excess Mn tolerance [96,97]. Mn poisoning stress increased the activity of POD and genetic expression of pea (*Pisum sativum*) and *S. guianensis*, both of which had a vital effect on plant acclimation to Mn toxicity [96,98]. In the present study, 2 *PODs* (*Glyma.18G055500* and *Glyma.17G198700*) were upregulated, whereas the other 12 *PODs* were downregulated in soybean roots (Table 2). Therefore, the upregulation or downregulation of the 14 *PODs* may be conducive to improving the tolerance of soybean roots to Mn poisoning.

Aux/IAA proteins are the negative regulation factors in the auxin signal transduction pathway [99]. The regulation of Aux/IAA homeostasis is regarded as the response of plants to external environment signals [64]. Mn poisoning decreases the content of auxin in plant roots by decreasing the biosynthesis of auxin and restraining the transport of auxin by reducing the gene expression of *PIN4* and *PIN7* belonging to the auxin efflux carrier family, and the findings of the qRT-PCR revealed that four auxin biological synthesis genes (*ASA1*, *SUR1*, *YUC2*, and *YUC3*) had observably reduced expression in responding to Mn poisoning [42]. In this research, the genetic expression of nine genes (*Glyma.08G010400*, *Glyma.06G134000*, *Glyma.02G142600*, *Glyma.03G029600*, *Glyma.05G101300*, *Glyma.09G011200*, *Glyma.19G258800*, *Glyma.17G046000*, and *Glyma.13G361200*) related to auxin synthesis was significantly downregulated (Table 1). The results implied that Mn poisoning stress could affect the genetic expression of genes in connection with auxin synthesis and thus affect the generation of growth in soybean roots.

Evidence showed that a gene belonging to the *bHLH* family of transcription factors, *AtNAI1*, affects the expression of *AtMEB1/2* and controls the tolerance of *Arabidopsis* to Mn poisoning [100]. In the present study, 94 transcription factors in soybean roots were found to have different responses to Mn toxicity. Of the 10 *bHLH transcription factors*, 6 (*Glyma.04G090100*, *Glyma.02G025500*, *Glyma.13G098000*, *Glyma.07G185300*, *Glyma.19G222000*, and *Glyma.18G039200*) were upregulated, whereas the other 4 (*Glyma.11G043700*, *Glyma.15G170500*, *Glyma.09G064200*, and *Glyma.01G197900*) were downregulated to adapt to Mn poisoning (Table 4). However, the mechanisms of the bHLH transcription factor family in the roots of soybean responding to Mn poisoning remain unclear. Hence, complicated adjusting mechanisms in soybean roots may be involved in responding to Mn toxicity, and these mechanisms need further exploration. 

The finding of the experiment demonstrated that after the soybean root system suffered high Mn poisoning, Mn toxicity could lead to increased oxidative stress and membrane damage by increasing the activities of superoxide free radicals and malondialdehyde (MDA), inducing downregulation of the expression levels of different DEGs, such as metal transport genes, hormone synthesis-related genes, and some transcription factors. The accumulation of metal ions, hormone synthesis, and the physiological and biochemical changes influenced the growth and metabolism of roots. In response to the toxicity stress of high Mn, the root cells activated the antioxidant oxidase system. Although the improvement in activity of antioxidant enzymes could relatively reduce the content of ROS, it could not reverse the trend of high ROS content in root cells under the toxicity stress of high Mn, resulting in irreversible cell damage. In conclusion, the toxicity stress of high Mn levels affected the cell integrity, nutrient uptake, hormone regulation, and resistance of soybean root growth. It resulted in inhibited root growth and development and then led to the decrease in root biological yield. The regulatory process of roots suffering Mn poisoning in soybean is shown in Figure 11. The results demonstrated that soybean roots may have complicated regulating mechanisms in responding to Mn poisoning stress, thus demanding further study.

## 4. Materials and Methods

### 4.1. Source of Plant Materials

The soybean variety used in this study is named YC03-3 (Yuechun 03-3), cultivated by the RBC (Root Biology Center in South China Agricultural University) and grown in the experimental base belonging to CCAS (College of Coastal Agricultural Sciences in Guangdong Ocean University) (eastern longitude: 110.300832, northern latitude: 21.151215). The seeds were germinated in sand and cultured for 8 days before being treated with Mn. As previously described, soybean seedlings with consistent growth were transported to a plastic box with a volume of 15 L for hydroponics, and improved Hoagland nutrient solution was added [2]. The nutrient solution used contained 400 µM NH_4_NO_3_, 25 µM MgCl_2_, 1500 µM KNO_3_, 1.5 µM ZnSO_4·_7H_2_O, 500 µM MgSO_4·_7H_2_O, 40 µM Fe-EDTA(Na), 300 µM K_2_SO_4_, 1200 µM Ca(NO_3_)_2_·4H_2_O, 300 µM (NH_4_)_2_SO_4_, 0.5 µM CuSO_4_·5H_2_O, 500 µM KH_2_PO_4_, 2.5 µM NaB_4_O_7_·10H_2_O, and 0.16 µM (NH_4_)_5_MoO_24_·4H_2_O. All of the chemical reagents used in this study were analytical grade (Solarbio corporation, Beijing, China). Meanwhile, 5, 50, 100, 150, 200, and 300 µM MnSO_4_ (Guangzhou Reagent, Guangzhou, China) was added into the nutrient solution for Mn treatment. The control group was treated with 5 µM Mn. The experiments were carried out in four biological replicates for every Mn concentration. A temperature range at 25–30 °C/18–22 °C was adopted to adjust the growth of the plant. The light cycle was about 12 h/day, and the culture solution was updated every 5 days. The pH value of the culture solution was regulated to 5.0 by 1 M potassium hydroxide or sulfuric acid every 2 days (Solarbio corporation, China). After the soybeans were treated with 5, 50, 100, 150, 200, and 300 µM Mn for 15 days, the roots were harvested to test their fresh weight, dry weight, growth indices, and various ion contents.

### 4.2. Surveying Dry and Fresh Biomass of Roots in Soybean

The fresh biomass of roots in soybean was measured immediately after harvest. The soybean roots were put in a constant-temperature oven (Shanghai Yiheng Corporation, Shanghai, China) and thoroughly dried at the temperature of 105 °C for 30 min. Therewith, the samples were then kept in an oven at 65 °C for 8 days. Afterwards, the dry weight was calculated [11]. There were four biological replicates for each treatment.

### 4.3. Assessment of Root Phenotype Data

The WinRHIZO technique was adopted to assess the root phenotype data of soybean seedlings as reported earlier [101]. Soybean roots from various experimental groups were acquired and fully spread on the image scanner (Epson, Tokyo, Japan). Test materials were assessed with the assistance of the software of WinRHIZO (WinRHIZO 2013e Professional Edition, WinRhizo Pro, Quebec, QC, Canada).

### 4.4. Determination of Content of MDA 

After soybean was treated with 5 (normal level of Mn) and 100 (Mn poisoning concentration) µM MnSO_4_ for 15 days, the physiological indexes of the roots were determined. The MDA content was measured by applying the improved technologies of TBA (thiobarbituric acid) [102]. In short, 0.1 g of the root was triturated and homogenized in 10 mL phosphate buffer solution (Guangzhou Reagent, Guangzhou, China, 0.05 M) and then isolated with 2 mL of 0.6% TBA (Guangzhou Reagent, Guangzhou, China). The extraction substance was put in a 100 °C constant-temperature water bath (Lichen corporation, Shanghai, China) for 15 min and then quickly cooled down with ice. After centrifugation was performed at the speed of 4200 rpm in a supercentrifuge (Eppendorf 5415D, Hamburg, Germany) for 20 min, an ultraviolet spectrophotometer (Yuanxi UV-5100B, Shanghai, China) was used. The absorbance of the liquid supernatant was determined at wavelengths of 450, 532, and 600 nm separately. The TBA compounds in MDA were quantified with the extinction coefficient.

### 4.5. Determination of concentration of soluble protein 

The Coomassie brilliant blue (CBB) method was accepted to test the content of soluble protein [103]. Root tissue homogenates (0.1 g) were extracted in 10 mL PBS (phosphatic buffer solution) (0.05 M, pH = 7.8), followed by 2.9 mL mixed solution including 0.1 g CBB G-250 (Guangzhou Reagent, Guangzhou, China). After the reaction for 2 min, the absorbency of the liquid supernatant was tested at a wavelength of 595 nm to compute soluble protein contents in samples by adopting the BSA (bovine serum albumin) standard curve (Guangzhou Reagent, Guangzhou, China).

### 4.6. Determination of Soluble Sugar Content

The anthrone method was used to determine the content of soluble sugar [104]. First, 0.2 g of fresh soybean root samples was weighed and mixed, placed in a test tube with a plug, and 15 mL distilled water was added. The compound was heated to boiling for 20 min, and the samples were removed, cooled, and filtered through a bottle with a capacity of 100 mL. Next, 1.0 mL of the extraction liquid to be measured was mixed with 5 mL anthrone reagent (Solarbio corporation, China) for extraction. The absorbency was determined at a wavelength of 620 nm. The standard curve of glucose (analysis of pure anhydrous glucose) was accepted to compute the concentration of sugar in samples.

### 4.7. Assaying the Content of Pro 

Root materials (0.1 g) were blended with 3% sulfosalicylic acid (10 mL) (Guangzhou Reagent, Guangzhou, China) and then filtered to assess the Pro concentration of soybean roots [105]. The reaction mixed solution consisted of 2 mL liquid supernatant, 2 mL glacial acetic acid (Solarbio corporation, China), and 3 mL acidic indole solution (Guangzhou Reagent, Guangzhou, China). The mixture was transferred to a glass tube, reacted at the temperature of 100 °C for 60 min, and cooled with an ice block. The product was isolated with 5 mL methylbenzene (Ghtech, Shantou, China) and rotated for 30 s. The color change was determined by an ultraviolet spectrophotometer at a wavelength of 520 nm, with methylbenzene as the blank control group. The calibration curve on account of the Pro standard was established to determine the Pro content in roots.

### 4.8. Enzyme Activity Measurement

Root tissues (0.1 g) treated with different concentrations of Mn were adequately crushed and blended with precooled phosphate buffers (0.05 M, pH = 7.8). And then, centrifugation was conducted at the temperature of 4 °C and velocity of 10,000 rpm for 20 min. The enzyme activity was measured in a supernatant solution. The vitality of SOD was surveyed by application of the means reported earlier [106]. The analysis system consisted of 0.5 mL botanical extracts and 1 mL 125 mM natrium carbonicum (Solarbio corporation, China), 0.4 mL 25 µM NBT (Nitro blue tetrazolium) (Solarbio corporation, Shanghai, China), and 0.2 mL 0.1 mM ethylenediamine tetraacetic acid (Ghtech, Shantou, China). Then, 0.4 mL 1 mM oxammonium hydrochloride (Guangzhou Reagent, Guangzhou, China) was supplemented to activate chemical reactions, and the light absorption value was surveyed at the wavelength of 560 nm. The SOD unit was represented by the quantity of enzyme requested to prevent a 50% drop in NBT. 

The activity of POD was surveyed as previously reported [107]. The mixture consisted of 30% hydrogen peroxide (Guangzhou Chemical, China) and 1% methyl catechol (Solarbio corporation, China), with a total volume of 3 mL. In reactions, 40 µL of the enzyme extract was supplemented into the compound. The absorbance variation due to oxidation of methyl catechol was surveyed at a wavelength of 470 nm. The activity of the POD unit was represented as a 0.01 decrease in OD470 nm within 1 min.

The CAT activity was determined using the reported technique [108]. The mixed substance contained 2.9 mL of 30% hydrogen peroxide, 0.1 mL enzyme extracted material, and 0.15 M PBS (pH = 7.0). The activity of CAT was evaluated by supervisory controlling the attenuation level of the absorbance value of hydrogen peroxide in OD240 nm.

The vitality of APX was measured via inspecting the absorbance of the oxygenization rate of ascorbate in OD290 nm [109]. The mixed solvent of the reaction was composed of 0.1 mL extract, 2.6 mM EDTA, 0.15 mM H_2_O_2_, and 0.15 mM ascorbic acid (Ghtech, China). The APX activity unit was expressed by the amount of enzymes requested to oxidize 1 µM of ascorbic acid.

### 4.9. Determination of Ion Content in Soybean Roots

Under different conditions of Mn treatment, soybean roots were taken on day 15 to determine their ion concentration. After the roots were dried and crushed, 0.2 g of samples was weighed, moved to a digestion tank (Changyi, KH-15, Beijing, China), and soaked overnight with 5 mL 98% sulfuric acid (Kermel, Tianjin, China). The digestion tank was placed in a constant-temperature oven (Beijing Yiheng BGP9050AH, Beijing, China) at a temperature of 80 °C for 2 h, at a temperature of 120 °C for 2 h, and at a temperature of 160 °C for 4 h, when the root materials were digested in transparent liquid. After the root materials were cooled down to 25 °C, the inner tank and lid of the digestion tank were cleaned three times with 1% sulfuric acid. The eluant was then moved to a measuring bottle with capacity of 50 mL (Robinde, Nanchang, China), and 1% sulfuric acid solution was added into the bottle until the scale line. The concentrations of Mg, Fe, and Mn in the roots of soybean were measured with an ICP-AES (inductively-coupled plasma atomic emission spectrometer) (Hitachi PS7800, Tokyo, Japan), with vacant digestive juices as the control group [110]. Elements were measured four times. The unique spectral wavelength of the elements was used to identify the types of metal elements, and the element contents were quantitatively analyzed by comparing the intensity of the mass spectrometry signal with the concentration of these elements.

### 4.10. Identification of the Presence of Four Different Hormones in the Roots of Soybean 

Phytohormones were acquired from the roots of soybean that dealt with various concentrations of Mn (5 and 100 μM MnSO_4_), and the hormone contents, such as indole butyric acid (IBA), indoleacetic acid (IAA), GA_3_, and ABA, in the soybean roots was determined using HPLC (high-performance liquid chromatography) (AGLIENT1290, Santa Clara, CA, USA) combined with series MS/MS (mass spectrometry) (AB SCIEX-6500Qtrap, San Diego, CA, USA); the internal standard substances were added into the extracts to correct the assay results [111,112]. External standards (ABA, IAA, IBA, and GA_3_) were pure chromatographic products (Sigma, Waltham, MA, USA). Deuterated IBA, IAA, GA_3_, and ABA (Sigma, USA) were used as internal standards. C18 QuECherS (Shanghai Amperex, China) was utilized to pack the chromatographic columns, and the acetonitrile and methyl alcohol were chromatographically pure (Merck, Darmstadt, Germany). The methods of hormone extraction and content measurement refer to those previously reported [111,112]. 

The produced standard curve working solution may be utilized to construct the standard curve for further hormone content computation. First, 988 μL of methanol solution (Kermel, China) was placed into 1.5 mL centrifuge tubes, and 2 L of each 500 g/mL of the corresponding hormone external standard stock solution was added, shaken well, and configured as the external standard master solution with an eventual concentration of 1 μg/mL. Then, 990 μL methyl alcohol solution was supplemented in a 1.5 mL centrifugal tube, and 2 L of each 500 μg/mL of the appropriate hormone internal standard stock solution was added and agitated vigorously to make the master batch of internal standard at a final concentration of 1 μg/mL. Finally, methanolic solutions were used to make standard curves with final concentrations of 0.1, 0.2, 0.5, 2, 5, 20, 50, and 200 ng/mL, with each concentration point containing 20 ng/mL of the appropriate hormone internal standard. On the basis of the results of the HPLC–MS/MS measurements, the standard curve can be plotted, where the horizontal coordinate X is the concentration of the external standard divided by the corresponding internal standard, and the vertical coordinate Y is the peak area of the external standard divided by the corresponding internal standard.

Four hormones were extracted from soybean roots by crushing samples to be analyzed in liquid nitrogen and carefully weighing 1 g of the samples in glass test tubes. Each hormone test was performed four times. A 10-fold amount of acetonitrile solution was added, as well as 8 μL of the 20 ng/mL master solution of the matching internal standard. The root samples were carefully extracted overnight at a temperature of 4 °C and centrifuged for 5 min at a speed of 12,000× *g* rpm. After the liquid supernatant was obtained, the deposit was treated with a fivefold amount of acetonitrile solution, extracted two times, and combined with the supernatants. The sample received about 35 mg of C18 packing before being shocked severely for 30 s and centrifuged at a speed of 10,000 rpm for 5 min. The liquid supernatant was then removed. The sample was blow-dried with nitrogen, redissolved in 400 μL methyl alcohol, filtered through a 0.22 μm millipore plastic membrane filter, and reserved at −20 °C in a refrigerator (Ronshen, China) for HPLC–MS/MS. Appendix A show the gradient parameters of HPLC and MS and the monitoring settings of the screening reaction for protonated or deprotonated plant hormones.

### 4.11. Library Preparation and Transcriptome Sequencing Analysis

Root samples were obtained for total RNA extraction and mRNA library formation of soybean seedlings cultured in hydroponic nutrient solution supplemented with 5 (control group) or 100 (Mn toxicity stress group) µM Mn for 15 days. Transcription sequencing was performed in accordance with a previously reported method [8]. Three biological replications were implemented for each sample. The ultrapure RNA kit (CWBIO, Jiangsu, China) was adopted for RNA extraction. Then, fragments and inverse transcription were performed by using random primers. Meanwhile, the entire library was amplified by PCR. The library was sequenced by application of the Illumina platform. Raw sequencing results were filtered to generate quality advanced data (clean data). Then, the advanced data were compared with the referenced gene sequence of soybean (NCBI) via tophat version 2.0.12 [113,114].

In order to quantify the expression of genes, HTSeq version 0.6.0 and transcription snippets per thousand bases per million readings were used. DESeq1.16 was used for the determination of differentially expressed genes, and Qlog2Ratio was used for calculating gene expression level [115]. The data were supplemented with the integrated gene expression database, with entry number PRJNA946643 (https://dataview.ncbi.nlm.nih.gov/object/PRJNA946643?reviewer=oh3htrv4ofv91c6e4jhe3t4poc (accessed on 20 March 2023). DAVID was adopted to implement GO function enrichment studies [116,117].

### 4.12. qRT-PCR Detection

Ribonucleic acid (RNA) was isolated from soybean roots using the ultrapure RNA kit (CWBIO, China). After genome DNA was removed, cDNA was produced using the reverse transcription kit (Takara, Maebashi, Japan). A real-time qPCR (Bio-rad Company, Hercules, CA, USA) was used to carry out qRT-PCR analysis, according to an early report [8]. In brief, the sample was diluted 30 times as a template, and the reaction procedure was as shown below: 95 °C for 30 s, 95 °C for 5 s, 60 °C for 15 s, and then 72 °C for 30 s. As the control group, the reference gene named *GmEF* (*Glyma.17G186600*) was used to calculate the relative transcription levels on the basis of the rate of the selected gene to the internal control gene, as previously reported [118]. Appendix A lists the primers adopted for qRT-PCR.

### 4.13. Data Analysis

Statistical package for the social sciences (SPSS) 17.0 (SPSS, USA) and Microsoft Excel 2010 (Microsoft, Redmond, WA, USA) were used for statistical analysis. Student’s *t*-test and Duncan’s multiple comparison test were used for statistical comparison and significant analysis [11,118].

## 5. Conclusions

The effects of Mn poisoning on soybean roots were examined at both the physiological and molecular levels in this study. Mn poisoning altered the internal and external osmotic equilibrium of the soybean root system, resulting in a substantial accumulation of Mn in the root. Furthermore, excessive Mn accumulation increased the activity of antioxidant enzymes in soybean roots, interfered with metal ion uptake and transport, as well as hormone synthesis, and disrupted root morphology, all of which suppressed soybean root growth and development. In addition, transcriptome sequencing revealed 1430 DEGs in soybean roots under Mn poisoning, of which 572 DEGs were upregulated and 858 DEGs were downregulated; and further, qRT-PCR was performed to test 30 of these DEGs (including related genes of ion-transporting proteins, hormones, antioxidant enzymes, and transcription factors), and the validation results were consistent with the transcriptome sequencing results. These findings deepen the cognition of the response of soybean roots to Mn poisoning and lay the foundation for further exploration of specific molecular regulatory mechanisms.

## Figures and Tables

**Figure 1 ijms-24-12727-f001:**
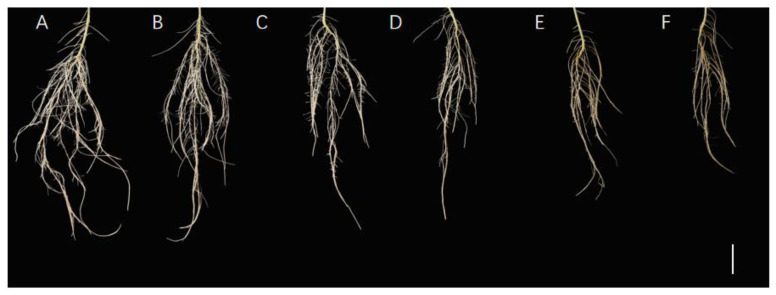
Influences of various concentrations of Mn on soybean root morphology. Root phenotypes of soybean with different Mn concentrations (from left to right: (**A**–**F**): 5, 50, 100, 150, 200, and 300 µM) after 15 days of treatment (bar = 3 cm).

**Figure 2 ijms-24-12727-f002:**
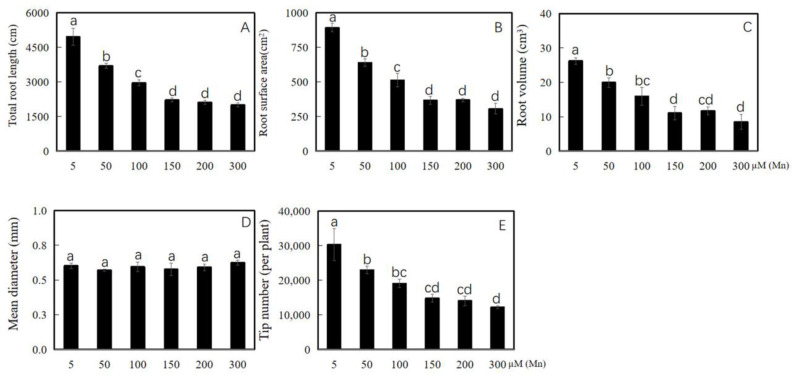
Results of various concentrations of Mn on growth of soybean root. (**A**) Total root length; (**B**) root area index; (**C**) root volume; (**D**) root mean diameter; and (**E**) root tips number. The data were represented by the mean value and standard deviation (n = 4). Duncan’s multiple comparison test was adopted for significant analysis of the differences in root growth between the normal group and the treatment group, and different letters on the bar chart indicated significant differences (*p* < 0.05).

**Figure 3 ijms-24-12727-f003:**
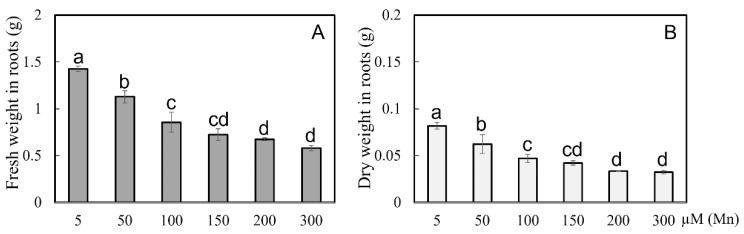
Results of various Mn concentrations on root weight of soybean. (**A**) Fresh and (**B**) dry weights of roots. The data are represented by the mean value and standard deviation (n = 4). Duncan’s multiple comparison test was adopted for significance analysis of the differences in root growth between the normal group and the treatment group, and different letters on the bar chart indicate significant differences (*p* < 0.05).

**Figure 4 ijms-24-12727-f004:**
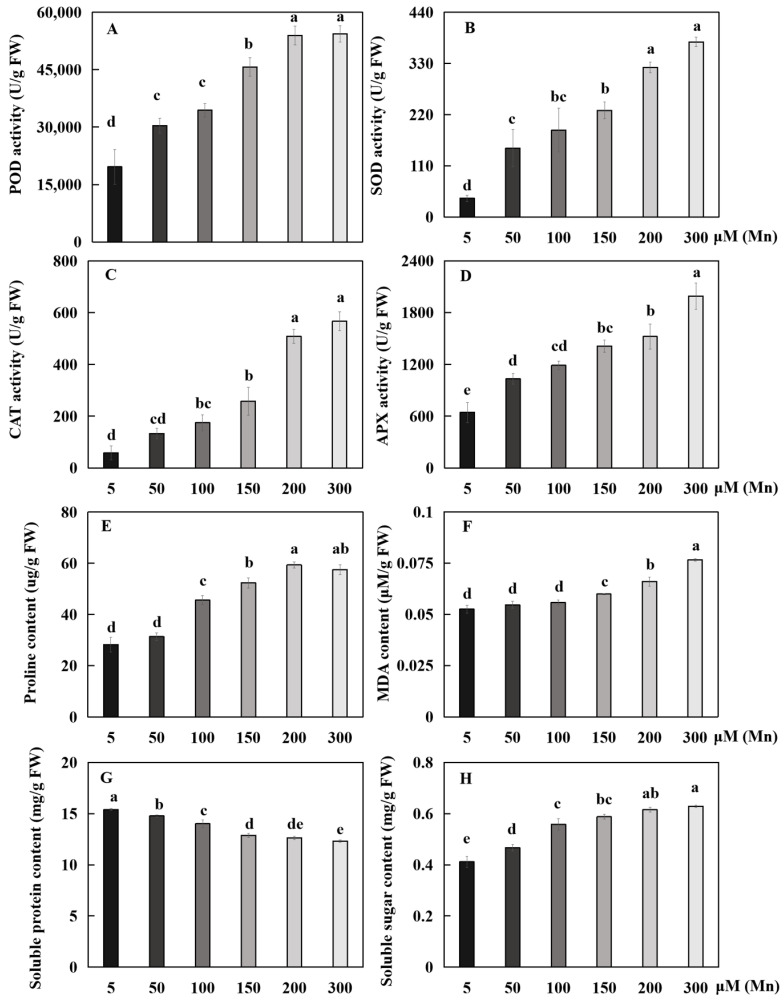
Results of various concentrations of exogenous Mn on activities of POD, SOD, CAT, and APX and contents of soluble protein, soluble sugar, Pro, and MDA in soybean root. Activity of (**A**) POD, (**B**) SOD, (**C**) CAT, and (**D**) APX; (**E**) content of Pro, (**F**) MDA, (**G**) soluble protein, and (**H**) soluble sugar. The data are represented by the mean value and standard deviation (n = 4). Duncan’s multiple comparison test was adopted for significance analysis of the differences in root growth between the normal group and the treatment group, and different letters on the bar chart indicate significant differences (*p* < 0.05).

**Figure 5 ijms-24-12727-f005:**
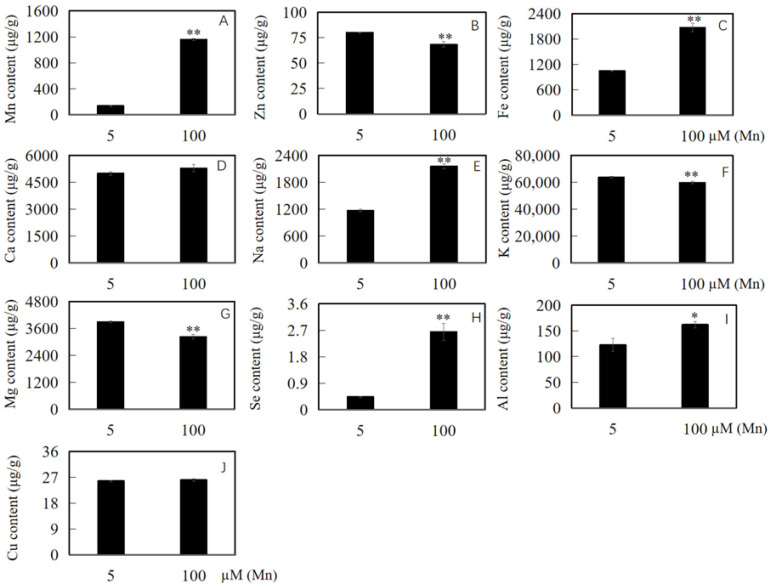
Results of various concentrations of Mn on ion accumulation in the roots of soybean. Content of (**A**) Mn, (**B**) Zn, (**C**) Fe, (**D**) Ca, (**E**) Na, (**F**) K, (**G**) Mg, (**H**) Se, (**I**) Al, and (**J**) Cu in soybean roots. The data are represented by the mean value and standard deviation (n = 4). The significance test of difference between normal group and Mn treatment group was implemented with Student’s *t* test at * *p* < 0.05 or ** *p* < 0.01.

**Figure 6 ijms-24-12727-f006:**
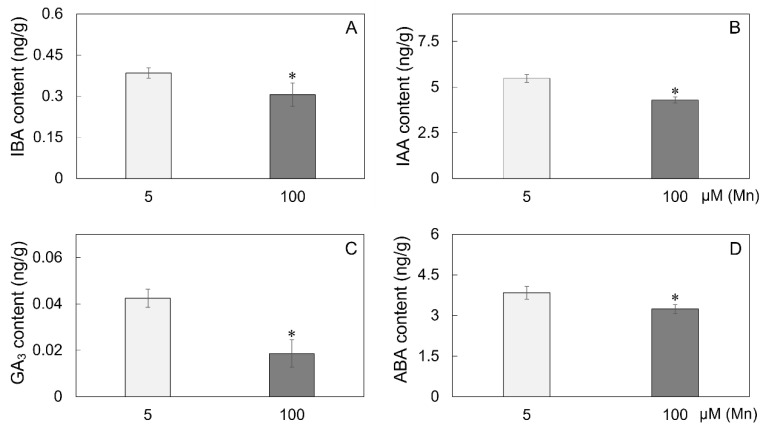
The content of hormones in the roots of soybean suffering Mn toxicity. Content of (**A**) indole butyric acid (IBA), (**B**) indoleacetic acid (IAA), (**C**) gibberellic acid 3 (GA_3_), and (**D**) abscisic acid (ABA) in soybean roots with 5 μM and 100 μM MnSO_4_ treatment. The data are represented by the mean value and standard deviation (n = 4). The significance test of difference between normal group and Mn treatment group was implemented with Student’s *t* test at * *p* < 0.05.

**Figure 7 ijms-24-12727-f007:**
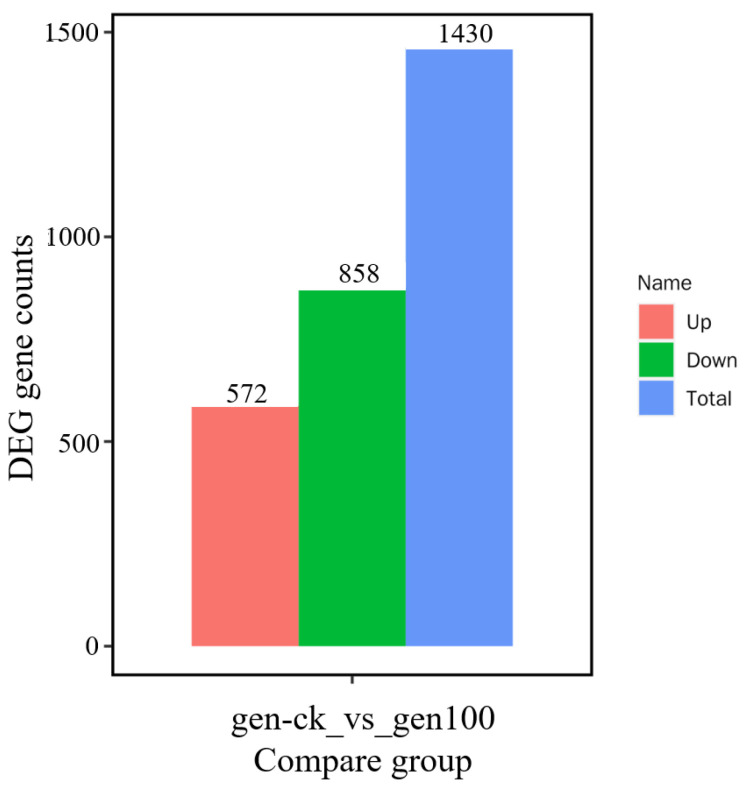
Statistical histogram of DEGs. Red represents the quantity of upregulation genes (572), green indicates the quantity of downregulation genes (858), and blue displays the total quantity of DEGs (1430).

**Figure 8 ijms-24-12727-f008:**
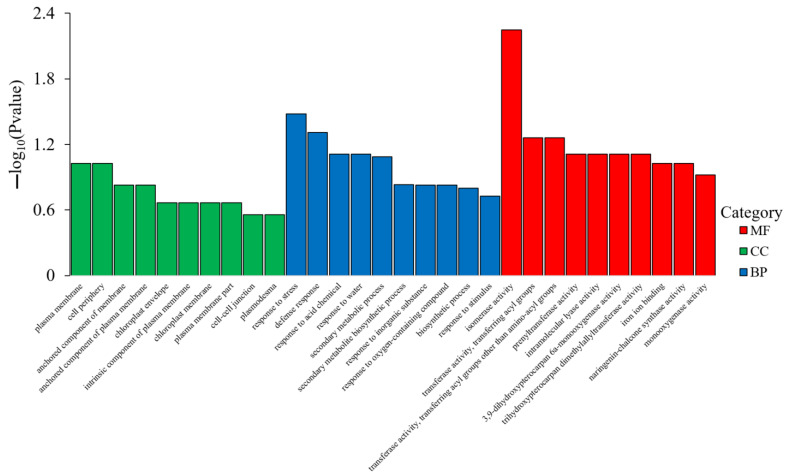
GO enrichment column of differential genes. MF: molecular function; BP: biological process; CC: cell composition. A maximum of ten GO entries with the smallest *p* value were selected for display.

**Figure 9 ijms-24-12727-f009:**
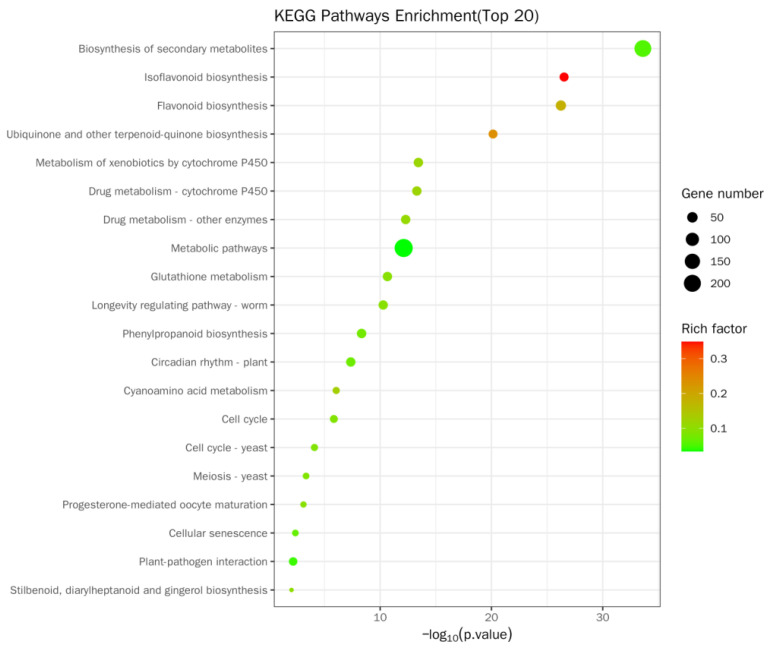
KEGG enrichment bubble map of DEGs. RF (Rich factor) means the percentage between the amount of DEGs situated in the passageway entry and the total quantity of genes situated in the passageway entry among all of the interpreted genes. The higher the RF is, the higher the gathering level. In general, the value scope of FDR is 0–1, and the nearer it is to zero, the more remarkable the enrichment is.

**Figure 10 ijms-24-12727-f010:**
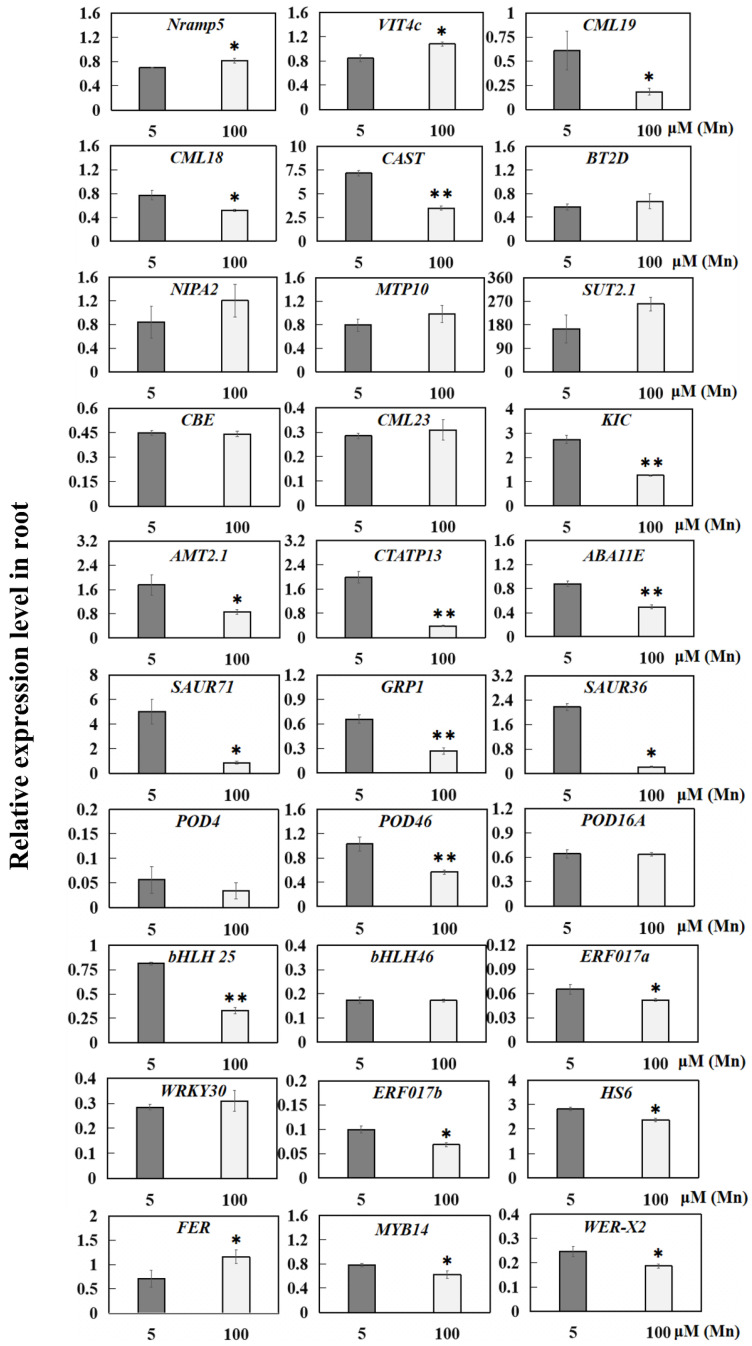
qRT-PCR results of 30 DEGs in the roots of soybean at 5 and 100 µM Mn concentrations. The relative expression level of DEGs in the roots of soybean. The data are represented by the mean value and standard deviation (n = 4). The significance test of difference between normal group and Mn treatment group was implemented with Student’s t test at * *p* < 0.05 or ** *p* < 0.01. *Nramp5*: *metal transporter*; *VIT4c*: *vacuolar iron transporter homolog 4*; *CBE, CML, CAST, and KIC*: *calcium-binding proteins*; *BT2D*: *boron transporter*; *NIPA2*: *magnesium transporter*; *MTP10*: *metal tolerance protein 10*; *SUT2.1*: *sulfate transporter 2.1*; *AMT2.1*: *ammonium transporter*; *CTATP13*: *calcium-transporting ATPase 13*; *ABA11E*: *ABA synthesis-related protein*; *SAUR*: auxin reactive protein; *GRP*: *gibberellin-responsive protein*; *POD*: *peroxidase; bHLH*: *bHLH transcription factor*; *ERF*: *ethylene response transcription factor; WRKY30*: *WRKY transcription factor*; *HS6*: *heat stress transcription factor*; *FER*: *FER transcription factor*; *MYB14*: *MYB transcription factor*; *WER-X2*: *WER transcription factor*.

**Figure 11 ijms-24-12727-f011:**
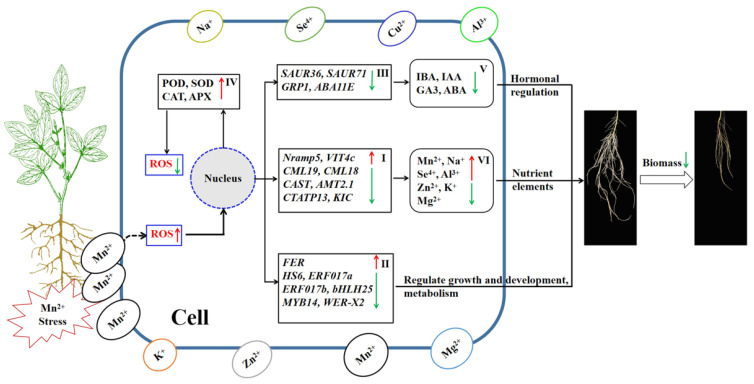
Regulatory pathway map of roots suffering Mn poisoning in soybean. I: ion transporter-related genes; II: transcription factors; III: hormone-related genes; IV: antioxidant enzymes; V: hormones; and VI: metal ions. Small red arrows express upregulated expression of genes, increased substance content, or increased enzyme activity. Small green arrows express downregulated expression of genes or reduced content of substances.

**Table 1 ijms-24-12727-t001:** DEGs identified as hormones.

Gene ID	log2FoldChange	Description
*Glyma.08G010400*	−5.2353	Auxin
*Glyma.06G134000*	−1.161	Auxin
*Glyma.02G142600*	−1.5184	Auxin
*Glyma.03G029600*	−2.5301	Auxin
*Glyma.05G101300*	−1.482	Auxin
*Glyma.09G011200*	−1.34	Auxin
*Glyma.19G258800*	−1.1334	Auxin
*Glyma.17G046000*	−1.5222	Auxin
*Glyma.13G361200*	−1.6535	Auxin
*Glyma.07G268400*	−3.3847	ABA
*Glyma.09G014700*	−1.4534	ABA
*Glyma.08G208500*	3.2515	Gibberellin (GA3)
*Glyma.12G216100*	−1.6498	Gibberellin (GA3)
*Glyma.13G259500*	3.4365	Gibberellin (GA3)
*Glyma.19G133600*	3.8827	Gibberellin (GA3)
*Glyma.16G200800*	2.2778	Gibberellin (GA3)
*Glyma.19G022500*	1.0633	Gibberellin (GA3)
*Glyma.12G137700*	−1.8625	Gibberellin (GA3)
*Glyma.09G149200*	1.2522	Gibberellin (GA3)
*Glyma.02G010100*	−1.2806	Gibberellin (GA3)
*Glyma.13G285400*	−2.3356	Gibberellin (GA3)
*Glyma.16G145100*	−1.6728	Salicylic acid (SA)

**Table 2 ijms-24-12727-t002:** DEGs identified as antioxidant enzymes.

Gene ID	log2FoldChange	Description
*Glyma.18G055500*	2.7191	Peroxidase
*Glyma.03G038200*	−3.4	Peroxidase
*Glyma.20G169200*	−1.7151	Peroxidase
*Glyma.16G055900*	−1.0025	Peroxidase
*Glyma.20G001400*	−1.2568	Peroxidase
*Glyma.01G130800*	−2.0236	Peroxidase
*Glyma.03G038700*	−1.2567	Peroxidase
*Glyma.01G130500*	−1.0805	Peroxidase
*Glyma.10G191700*	−3.9194	Peroxidase
*Glyma.09G022800*	−1.6039	Peroxidase
*Glyma.09G284400*	−1.7185	Peroxidase
*Glyma.02G234200*	−1.9256	Peroxidase
*Glyma.17G198700*	1.159	Peroxidase
*Glyma.14G104400*	−3.2773	Peroxidase

**Table 3 ijms-24-12727-t003:** DEGs identified as ion transporters.

Gene ID	log2FoldChange	Description
*Glyma.09G122600*	1.5918	Metal tolerance protein
*Glyma.18G168900*	1.8991	Sulfate transporter
*Glyma.02G082500*	1.301	Vacuolar iron transporter
*Glyma.08G076000*	1.5034	Vacuolar iron transporter
*Glyma.08G075900*	1.9643	Vacuolar iron transporter
*Glyma.05G121300*	1.7505	Vacuolar iron transporter
*Glyma.11G083300*	2.1146	Vacuolar iron transporter
*Glyma.09G031400*	1.335	Boron transporter
*Glyma.05G153000*	4.1071	Magnesium transporter
*Glyma.19G038600*	−1.4041	Calcium-transporting ATPase
*Glyma.07G153800*	−2.9733	Ammonium transporter
*Glyma.20G082450*	−1.0318	Ammonium transporter
*Glyma.19G199900*	2.257	Aluminum-activated malate transporter
*Glyma.06G115800*	2.2447	Metal transporter Nramp5
*Glyma.06G074100*	2.0172	Molybdate transporter
*Glyma.09G270900*	−2.1497	Calcium-binding protein
*Glyma.18G039500*	−1.2755	Calcium-binding protein
*Glyma.12G089800*	−1.6159	Calcium-binding protein
*Glyma.11G217200*	−1.4206	Calcium-binding protein
*Glyma.14G215800*	−1.1839	Calcium-binding protein
*Glyma.06G034700*	−1.2927	Calcium-binding protein
*Glyma.02G265900*	−1.4369	Calcium-binding protein
*Glyma.04G136600*	−2.9988	Calcium-binding protein
*Glyma.12G197900*	−1.8754	Calcium-binding protein
*Glyma.16G059300*	−1.6615	Calcium-binding protein
*Glyma.04G078400*	−1.0643	Calcium-binding protein
*Glyma.20G032500*	−2.3907	Phosphate transporter

**Table 4 ijms-24-12727-t004:** DEGs identified as transcription factors.

Gene ID	log_2_FoldChange	Description
*Glyma.13G267700*	−1.103	WRKY transcription factor
*Glyma.13G267600*	−2.0811	WRKY transcription factor
*Glyma.13G267500*	−1.1482	WRKY transcription factor
*Glyma.06G307700*	−1.314	WRKY transcription factor
*Glyma.03G002300*	−1.7427	WRKY transcription factor
*Glyma.08G320200*	−1.167	WRKY transcription factor
*Glyma.17G222300*	−2.2559	WRKY transcription factor
*Glyma.07G262700*	−1.8103	WRKY transcription factor
*Glyma.06G142100*	−2.6582	WRKY transcription factor
*Glyma.04G238300*	−3.1185	WRKY transcription factor
*Glyma.03G256700*	−1.8202	WRKY transcription factor
*Glyma.10G230200*	−1.2325	WRKY transcription factor
*Glyma.13G117600*	−1.4937	WRKY transcription factor
*Glyma.19G254800*	−2.9872	WRKY transcription factor
*Glyma.09G240000*	−1.3167	WRKY transcription factor
*Glyma.18G208800*	−1.1169	WRKY transcription factor
*Glyma.14G103100*	−2.5033	WRKY transcription factor
*Glyma.13G370100*	−2.2557	WRKY transcription factor
*Glyma.08G021900*	−2.9632	WRKY transcription factor
*Glyma.06G147100*	−1.0236	WRKY transcription factor
*Glyma.01G128100*	−2.4649	WRKY transcription factor
*Glyma.05G215900*	−2.6836	WRKY transcription factor
*Glyma.03G042700*	−2.4868	WRKY transcription factor
*Glyma.01G224800*	−2.7566	WRKY transcription factor
*Glyma.09G005700*	−1.0228	WRKY transcription factor
*Glyma.15G110300*	−1.0264	WRKY transcription factor
*Glyma.05G211900*	−1.2824	WRKY transcription factor
*Glyma.18G213200*	−1.7942	WRKY transcription factor
*Glyma.09G274000*	−1.832	WRKY transcription factor
*Glyma.06G142000*	−1.1276	WRKY transcription factor
*Glyma.16G026400*	−2.4819	WRKY transcription factor
*Glyma.06G125600*	−2.3697	WRKY transcription factor
*Glyma.17G224800*	−3.7504	WRKY transcription factor
*Glyma.07G057400*	−2.1102	WRKY transcription factor
*Glyma.18G238200*	−1.0434	WRKY transcription factor
*Glyma.04G223300*	−1.5064	WRKY transcription factor
*Glyma.06G061900*	−2.2066	WRKY transcription factor
*Glyma.11G043700*	−3.1903	bHLH transcription factor
*Glyma.15G170500*	−1.3616	bHLH transcription factor
*Glyma.04G090100*	1.9891	bHLH transcription factor
*Glyma.02G025500*	1.0493	bHLH transcription factor
*Glyma.13G098000*	1.8284	bHLH transcription factor
*Glyma.09G064200*	−1.4571	bHLH transcription factor
*Glyma.01G197900*	−2.2723	bHLH transcription factor
*Glyma.07G185300*	1.4333	bHLH transcription factor
*Glyma.19G222000*	1.1226	bHLH transcription factor
*Glyma.18G039200*	2.2407	bHLH transcription factor
*Glyma.10G010400*	−4.0787	MYB transcription factor
*Glyma.06G300000*	−2.5385	MYB transcription factor
*Glyma.05G234600*	−1.7641	MYB transcription factor
*Glyma.10G180800*	−1.3551	MYB transcription factor
*Glyma.12G104600*	−2.8527	MYB transcription factor
*Glyma.06G300200*	−3.1265	MYB transcription factor
*Glyma.19G218800*	−2.8762	MYB transcription factor
*Glyma.07G178500*	−1.0826	MYB transcription factor
*Glyma.20G209700*	−1.3008	MYB transcription factor
*Glyma.02G005600*	−1.2669	MYB transcription factor
*Glyma.03G221700*	−2.3185	MYB transcription factor
*Glyma.01G016600*	−1.0816	MYB transcription factor
*Glyma.13G228900*	−1.0593	MYB transcription factor
*Glyma.15G259400*	1.1537	MYB transcription factor
*Glyma.06G178600*	−1.8058	MYB transcription factor
*Glyma.06G300400*	−3.4385	MYB transcription factor
*Glyma.01G224900*	−1.4078	MYB transcription factor/CPC transcription factor
*Glyma.19G104201*	−1.8765	Ethylene-responsive transcription factor
*Glyma.02G067600*	2.3504	Ethylene-responsive transcription factor
*Glyma.13G227066*	−1.5663	Ethylene-responsive transcription factor
*Glyma.17G047300*	−3.9674	Ethylene-responsive transcription factor
*Glyma.10G016500*	−1.1183	Ethylene-responsive transcription factor
*Glyma.07G212400*	−3.0852	Ethylene-responsive transcription factor
*Glyma.14G020100*	−1.1127	Ethylene-responsive transcription factor
*Glyma.13G112400*	−3.563	Ethylene-responsive transcription factor
*Glyma.02G132500*	−3.2647	Ethylene-responsive transcription factor
*Glyma.16G047600*	−1.6488	Ethylene-responsive transcription factor
*Glyma.13G122766*	−4.1664	Ethylene-responsive transcription factor
*Glyma.13G088100*	−1.0779	Ethylene-responsive transcription factor
*Glyma.11G014200*	1.1517	Ethylene-responsive transcription factor
*Glyma.19G026000*	−1.8527	Ethylene-responsive transcription factor
*Glyma.10G119100*	1.2716	Ethylene-responsive transcription factor
*Glyma.15G077100*	−2.0076	Ethylene-responsive transcription factor
*Glyma.11G068700*	1.4964	GATA transcription factor
*Glyma.04G008900*	−1.0814	GATA transcription factor
*Glyma.03G135800*	−1.9691	Heat stress transcription factor
*Glyma.19G137800*	−1.1508	Heat stress transcription factor
*Glyma.19G132500*	1.5119	ORG transcription factor
*Glyma.06G114000*	−1.0913	NAC transcription factor
*Glyma.05G195000*	−1.1467	NAC transcription factor
*Glyma.12G178500*	2.7368	Iron deficiency induces transcription factors
*Glyma.06G090900*	−2.3686	TGA transcription factor
*Glyma.17G093600*	1.1858	E2FE transcription factor
*Glyma.12G104800*	−2.7972	WER transcription factor
*Glyma.08G169451*	−1.949	JUNGBRUNNEN transcription factor
*Glyma.17G231900*	−1.6796	RAX transcription factor
*Glyma.07G126900*	2.8666	RAX transcription factor

## Data Availability

Sequencing data generated in the study are available in the NCBI Sequence Read Archive (SRA) under BioProject accession PRJNA946643.

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
