# Peer review of "Transcriptome Sequencing Analysis of Root in Soybean Responding to Mn Poisoning"

_ijms, 2023, doi:10.3390/ijms241612727_

Round 1

Reviewer 1 Report

The manuscript written by Liu et al., 2023 entitled "Transcriptome Sequencing Analysis of Root in Soybean Re-sponding to Mn Poisoning" is of great significance in the field of stress biology and plant physiology. I have gone through the manuscript and found that this research work is scientifically sound, and presents great novelty. Overall, the English quality should be improved during the revision. Overall, I suggest a "Major revision" for this paper, and some of my comments are listed below.

1: Replace the following sentence "Mn is one kind of the essential trace elements for plant normal development, and excessive Mn can cause plant growth and development to be hindered", with "Manganese (Mn) is among one of the essential trace elements for normal plant development, however, excessive Mn can cause plant growth and development to be hindered."

2: Check for the sentences "starting with an abbreviation". Change to the proper name, at least at first mention.

3: The abstract is not comprehensive. It should be reduced in length by adding only very significant results and a solid conclusion.

4: Replace "Mn (Manganese)" to "Manganese (Mn), in Introduction".

5: At the end of the Introduction, the "Objectives" of the current research are not clearly presented and it seems "Results instead of objectives" are written here before making any hypothesis about the experiment. Please re-write the idea of the research clearly.

6: The results section seems OK to me.

7: Could the authors, please highlight the "mechanisms in discussion section" regarding the poisoning of Mn in soybean?

8: How the concentrations of Mn were chosen? was there any preliminary experiment conducted?

9: Why the soybean variety used in this study is named YC03-3 (Yuechun 03-3), was chosen?

10: Was Mn added via Hoagland nutrient media? how the dosage was controlled?

11: The conclusion should be rewritten with specifically emphasize on the study theme. 

Dear Editor;

The English language of the manuscript seems Ok to me. However, some typos and grammar were found and I have mentioned some of them in the comments.

regards 

kamran

Author Response

1: Replace the following sentence "Mn is one kind of the essential trace elements for plant normal development, and excessive Mn can cause plant growth and development to be hindered", with "Manganese (Mn) is among one of the essential trace elements for normal plant development, however, excessive Mn can cause plant growth and development to be hindered."

My response: L11-12. This is a good suggestion. I have modified it as requested.   

2: Check for the sentences "starting with an abbreviation". Change to the proper name, at least at first mention.

My response: L38, 244, 253, 346, 529, 544, 741, 877, 888. OK. That's pretty good advice. I have modified them as suggested.

3: The abstract is not comprehensive. It should be reduced in length by adding only very significant results and a solid conclusion.

My response: L11-33. This is a good advice. I have modified the abstract as suggested.

Manganese (Mn) is among one of the essential trace elements for normal plant development, however, excessive Mn can cause plant growth and development to be hindered. Nonetheless, the regulatory mechanisms of plant root response to Mn poisoning remains unclear. In the present study, results revealed that the root growth was inhibited, when exposed to Mn poisoning. Physiological results showed that the antioxidase enzymes activities (peroxidase, superoxide dismutase, ascorbate peroxidase, and catalase) and the proline, malondialdehyde, and soluble sugar contents increased significantly under Mn toxicity stress (100 μM Mn), whereas the soluble protein and four hormones (indolebutyric acid, abscisic acid, indoleacetic acid, and gibberellic acid 3) contents decreased significantly. In addition, the Mn, Fe, Na, Al, and Se contents in the roots increased significantly, whereas those of Mg, Zn, and K decreased significantly. Furthermore, RNA sequencing (RNA-seq) analysis was used to test the differentially expressed genes (DEGs) of soybean root under Mn poisoning. The results found 45,274 genes in soybean root and 1430 DEGs under Mn concentrations of 5 (normal) and 100 (toxicity) μM. Among these DEGs, 572 were upregulated and 858 were downregulated, indicating that soybean roots may initiate complex molecular regulatory mechanisms on Mn poisoning stress. The results of quantitative RT-PCR indicated that many DEGs were upregulated or downregulated markedly in the roots, suggesting that the regulation of DEGs may be complex. Therefore, the regulatory mechanism of soybean root on Mn toxicity stress is complicated. Present results lay the foundation for further study on the molecular regulation mechansim of function genes involved in regulating Mn tolerance traits in soybean roots.

4: Replace "Mn (Manganese)" to "Manganese (Mn), in Introduction".

My response: L38. This is a good suggestion. I have modified it as requested.

5: At the end of the Introduction, the "Objectives" of the current research are not clearly presented and it seems "Results instead of objectives" are written here before making any hypothesis about the experiment. Please re-write the idea of the research clearly.

My response: L100-104. OK. I have modified it as suggested.

For the past few years, RNA sequencing (RNA-seq) has been widely used to explore the mechanism of plants, including Citrus sinensis, Citrus grandis, Manihot esculenta, and Cucumis melo response to toxicity stress of Al, Cu, Fe, Ca, and other heavy metals [29-32]. Nevertheless, few studies have been reported on Mn poisoning in soybean, and the molec-ular regulation mechanism of Mn stress in soybean remains unclear. In the present study, high-throughput deep sequencing was used to conduct whole-genome transcriptome analysis of soybean roots and comparative analysis of Mn toxicity response genes in roots. In addition, the physiological response indices and the contents of various metal ions in roots treated with different concentrations of Mn were studied. The findings revealed fur-ther information about regulating specific signaling pathways that participate in adjusting the root resistance to Mn poisoning. They may offer preliminary foundation for further study of genes functions of resistance to Mn poisoning. Through the above research methods, this study will explore the molecular mechanism of soybean root response to Mn poisoning, further discover some key genes playing regulatory roles in Mn toxicity stress, and lay a foundation for cultivating high-quality soybean varieties with Mn toxicity tolerance traits.

6: The results section seems OK to me.

My response: Thank you for your recognition and affirmation.

7: Could the authors, please highlight the "mechanisms in discussion section" regarding the poisoning of Mn in soybean?

My response: L657-681. OK. I have modified it as suggested.

The finding of the experiment demonstrated that after the soybean root system was suffered to high Mn poisoning, Mn toxicity could lead to increased oxidative stress and membrane damage by increasing the activities of superoxide free radicals and malondialdehyde (MDA), inducing downregulation of the expression levels of different DEGs, such as metal transport genes, hormone synthesis-related genes, and some transcription factors. The accumulation of metal ions, hormone synthesis, and the physiological and biochemical changes influenced the growth and metabolism of roots. In response to the toxicity stress of high Mn, the root cells activated the antioxidant oxidase system. Although the improving in activity of antioxidant enzyme could relatively reduce the content of ROS, it could not reverse the trend of high ROS content in root cells under the toxicity stress of high Mn, resulting in irreversible cell damage. In conclusion, the toxicity stress of high Mn affected the cell integrity, nutrient uptake, hormone regulation, and resistance of soybean root growth. It resulted in inhibited root growth and development and then led to the decrease in root biological yield. The regulatory process of root suffering Mn poisoning in soybean was shown in Figure 11. The results demonstrated that soybean roots may have complicated regulating mechanisms in responding to Mn poisoning stress, thus demanding to study further.

8: How the concentrations of Mn were chosen? was there any preliminary experiment conducted?

 My response: These Mn treatment concentrations we used were not randomly chosen, but were screened through preliminary experiments.

9: Why the soybean variety used in this study is named YC03-3 (Yuechun 03-3), was chosen?

 My response: This soybean variety named Yuechun 03-3 (YC03-3) is a soybean variety that I have been using since I was a postdoc (2017-2019), and our lab propagates this soybean seed every year.

10: Was Mn added via Hoagland nutrient media? how the dosage was controlled?

 My response: Yes, different concentrations of MnSO4 were added to the modified Hoagland nutrient solution (Mn free). In this study, Mn mother liquor was first prepared, and then the appropriate amount of mother liquor was used and diluted to different final concentrations by calculation.

11: The conclusion should be rewritten with specifically emphasize on the study theme. 

My response: L893-908. OK. I have modified the conclusion as suggested.

The effects of Mn poisoning on soybean root were examined at both the physiological and molecular levels in this study. Mn poisoning altered the internal and external osmotic equilibrium of the soybean root system, resulting in a substantial accumulation of Mn in the root. Furthermore, excessive Mn accumulation increased the activity of antioxidant enzymes in soybean root, interfered with metal ion uptake and transport, as well as hormone synthesis, and disrupted root morphology, all of which suppressed soybean root growth and development. In addition, transcriptome sequencing revealed 1430 DEGs in soybean root under Mn poisoning, of which 572 DEGs were up-regulated and 858 DEGs were down-regulated; and further, qRT-PCR was performed to test 30 of these DEGs (including related genes of ion-transporting proteins, hormones, antioxidant enzymes, and transcription factor), and the validation results were consistent with the transcriptome sequencing results. These findings deepen the cognition of the responding of soybean root to Mn poisoning and lay the foundation for further exploration of specific molecular regulatory mechanisms.

Reviewer 2 Report

Dear Authors,

Congratulations on your work.

In this manuscript, the authors investigated the regulatory mechanisms of soybean root response to Mn poisoning. The topic is of interest but there are several issues that should be addressed before I can indicate this MS for publication.

Also, the MS should have line number to simplify the revision process. Please take this into consideration before submitting the improved version.

Figures captions should also be carefully checked and corrected. There are several font sizes which should be homogenized. 

The discussion section should be improved, and the results should be better discussed. For instance, the authors should correlate the results from the antioxidant enzyme activity or the hormones quantification with the RNA-seq and qRT-PCR analysis, instead of simply debiting literature

I think that the MS has potential to published, but first the authors should properly revise all the document.

Specific comments:

Keywords: should be provided in alphabetical order

Introduction

These enzymes play key roles in respiratory action, photosynthetic photosynthesis, protein and hormone synthesis in plants [1].” ->  photosynthetic photosynthesis? Please rewrite.

Although Mn is a kind of the essential trace elements for plant normal development, maybe excess Mn harmful to plant [3].” -> Please rewrite as grammatically the sentence makes no sense.

“...the accumulating of ROS (active oxygen)” -> Please change “accumulating” to “accumulation” and “(active oxygen)” to “(reactive oxygen species)”

Results:

The total root length significantly reduced by 25.49%, 40.35%, 55.32%, 57.26%, and 59.69%, respectively (Figure 2A).” The verb is missing in this sentence and in several other, i.e., “The total root length significantly reduced” should be “The total root length was significantly reduced”. Please do the same correction whenever needed. Also, there is no need to use two decimal places. Please change it to one. Same comment is valid for other paragraphs/sections.

Figure 2: “The data were represented by the mean value and standard deviation (SD) from repeating four times.”. Does this mean that your number of replicates was 4? If yes, please simplify it to “The data were represented by the mean value and standard deviation (n=4)”. Same comment valid for other figures. What do you mean with “normal control”? Normal Mn concentration? I don’t think that call it “normal control” is ok. Actually, this is simply the control.

Figure 2: “and various letters suggesting significance of difference (P < 0.05)”. Please rewrite.

“The physiological indices [POD, SOD, CAT, and ascorbate peroxidase (APX)] of soybean root were subjected to the stress response of Mn poisoning.” Please rewrite the sentence as it is a bit strange to say that “the physiological indices were subjected to the stress response of Mn poisoning”. Also, I think that is better to call it antioxidant enzymes instead of physiological indices, as this is a quite vague name.

Figure 4: POD, SOD, CAT, APX, Pro, and MDA should be defined in the figure’s caption.

Figure 7: Red represents the quantity of up-regulation genes, green indicates the quantity of down-regulation genes, and blue displays the total quantity of DEGs. -> where we read up-regulation genes, we should read up-regulated genes. This comment is also valid for when you are describing these results.

The GO (gene ontology) concentration of DEGs in soybean roots was analyzed in accordance with BP, CC, and MF.” Please write the meaning of BP. CC and MF.

The finding of the experiment demonstrated that after the soybean root system was suffered to high Mn poisoning, Mn toxicity could lead to increased oxidative stress and membrane damage by increasing the activities of superoxide free radicals and malondial- dehyde (MDA), inducing downregulation of the expression levels of different DEGs, such as metal transport genes, hormone synthesis-related genes, and some transcription fac- tors. The accumulation of metal ions, hormone synthesis, and the physiological and bio- chemical changes influenced the growth and metabolism of roots. In response to the tox- icity stress of high Mn, the root cells activated the antioxidant oxidase system. Although the improving in activity of antioxidant enzyme could relatively reduce the content of ROS, it could not reverse the trend of high ROS content in root cells under the toxicity stress of high Mn, resulting in irreversible cell damage. In conclusion, the toxicity stress of high Mn affected the cell integrity, nutrient uptake, hormone regulation, and resistance of soybean root growth. It resulted in inhibited root growth and development and then led to the decrease in root biological yield. The regulatory process of root suffering Mn poisoning in soybean was shown in Figure 11. The results demonstrated that soybean roots may have complicated regulating mechanisms in responding to Mn poisoning stress, thus demanding to study further.” -> To me this is clearly discussion section, so I suggest move it to there.

Discussion:

General comment: Please mention the relevant figures whenever you are describing your results. It makes easier to the reader to observe what you are discussion

“Excessive effective Mn”. Please clarify.

“APX, POD, and SOD are the primary antioxidant substances”. APX, POD and SOD are not antioxidant substances, they are antioxidant enzymes!

Material and Methods

4.1 Section. Was the EC of the nutrient solution controlled?

4.2 Section: “and thoroughly dried at the temperature of 105 °C for maintaining 30 min”. Are the authors sure that 30 min were enough to have the dry weight?

Each experiment was repeated four times” Please specify what do you mean with this sentence. Did you perform 4 experiments? Or did you have 4 biological replicates?? Please clarify it.

The CAT activity was determined using the reported technique [108]. The mixed sub- stance contained 2.9 mL of 30% hydrogen peroxide, 0.1 mL enzyme extracted material, and 0.15 M PBS (pH=7.0). The activity of CAT was evaluated by supervisory controlling the attenuation level of the absorbance value of hydrogen peroxide in OD470 nm.” Why did the authors measured Cat activity at 470 nm if the usual absorvance is 240 nm, including in the reference that the authors mention. Please clarify it.  Please also clarify if kinetics was used (and how – time range) to quantify CAT and APX activity.

Please briefly described the extraction method for hormone quantification.

There are some english issues that must be fixed before publication

Author Response

  1. Keywords:should be provided in alphabetical order

 My response: L34. That's pretty good advice. I have modified it as suggested.

Manganese poisoning; Root system; Soybean; Transcription analysis.

2.Introduction

“These enzymes play key roles in respiratory action, photosynthetic photosynthesis, protein and hormone synthesis in plants [1].” ->  photosynthetic photosynthesis? Please rewrite.

 My response: L40. OK. I have modified it as suggested.

The “photosynthetic photosynthesis” has been changed with “photosynthesis”.

3.“Although Mn is a kind of the essential trace elements for plant normal development, maybe excess Mn harmful to plant [3].” -> Please rewrite as grammatically the sentence makes no sense.

My response: L41-42. OK. I have modified it as suggested.

Although Mn is one of the important trace elements for plant development, excessive Mn may be harmful to plants.

4.“...the accumulating of ROS (active oxygen)” -> Please change “accumulating” to “accumulation” and “(active oxygen)” to “(reactive oxygen species)”

 My response: L84. OK. I have modified them as suggested.

The “accumulating” has been changed into “accumulation”, and “active oxygen” has been changed into “reactive oxygen species”.

5.Results:

 “The total root length significantly reduced by 25.49%, 40.35%, 55.32%, 57.26%, and 59.69%, respectively (Figure 2A).” The verb is missing in this sentence and in several other, i.e., “The total root length significantly reduced” should be “The total root length was significantly reduced”. Please do the same correction whenever needed. Also, there is no need to use two decimal places. Please change it to one. Same comment is valid for other paragraphs/sections.

My response: L115, 117, 118, 120, 140, 141, 145, 162, 168, 169, 171, 174, 195, 196, 198. OK. I have modified them as suggested.

L116-117, 119-120, 142-144, 163-166, 168-170, 172-175, 195, 197-198, 217, 231, 233. Furthermore, I have changed all the experimental data to one decimal places.

6.Figure 2: “The data were represented by the mean value and standard deviation (SD) from repeating four times.”. Does this mean that your number of replicates was 4? If yes, please simplify it to “The data were represented by the mean value and standard deviation (n=4)”. Same comment valid for other figures. What do you mean with “normal control”? Normal Mn concentration? I don’t think that call it “normal control” is ok. Actually, this is simply the control.

My response: L131-134; L150-152; L181-184; L203-204; L222-224; L374-376. Yes, this means that each experiment is repeated four times. That's pretty good advice. I have modified them as suggested. The “normal control” has been changed into “normal group”.

7.Figure 2: “and various letters suggesting significance of difference (P < 0.05)”. Please rewrite.

My response: L135-136; L153-154; L184-186. OK. I have modified it as follows.

“and different letters on the bar chart indicated significant differences (P < 0.05).”

8.“The physiological indices [POD, SOD, CAT, and ascorbate peroxidase (APX)] of soybean root were subjected to the stress response of Mn poisoning.” Please rewrite the sentence as it is a bit strange to say that “the physiological indices were subjected to the stress response of Mn poisoning”. Also, I think that is better to call it antioxidant enzymes instead of physiological indices, as this is a quite vague name.

 My response: L157-158. That's pretty good advice. I have modified it as suggested.

The L157. The “physiological indices” has been changed into “antioxidant enzymes”.

9.Figure 4: POD, SOD, CAT, APX, Pro, and MDA should be defined in the figure’s caption.

  My response: L178-179. OK. I have modified them as suggested.

Figure 4. Results of various concentrations of exogenous Mn on activities of POD, SOD, CAT, and APX and contents of soluble protein, soluble sugar, Pro, and MDA in soybean root.

10.Figure 7: Red represents the quantity of up-regulation genes, green indicates the quantity of down-regulation genes, and blue displays the total quantity of DEGs. -> where we read up-regulation genes, we should read up-regulated genes. This comment is also valid for when you are describing these results.

 My response: L239-241. OK. I have modified them as suggested.

 Figure 7. Statistical histogram of DEGs. Red represents the quantity of up-regulation genes (572), green indicates the quantity of down-regulation genes (858), and blue displays the total quantity of DEGs (1430).

11.“The GO (gene ontology) concentration of DEGs in soybean roots was analyzed in accordance with BP, CC, and MF.” Please write the meaning of BP. CC and MF.

My response: L245-246. OK. I have written the meaning of BP, CC and MF.

The gene ontology (GO) concentration of DEGs in soybean roots was analyzed in accordance with biological process (BP), cell composition (CC), and molecular function (MF).

12.“The finding of the experiment demonstrated that after the soybean root system was suffered to high Mn poisoning, Mn toxicity could lead to increased oxidative stress and membrane damage by increasing the activities of superoxide free radicals and malondial- dehyde (MDA), inducing downregulation of the expression levels of different DEGs, such as metal transport genes, hormone synthesis-related genes, and some transcription fac- tors. The accumulation of metal ions, hormone synthesis, and the physiological and bio- chemical changes influenced the growth and metabolism of roots. In response to the tox- icity stress of high Mn, the root cells activated the antioxidant oxidase system. Although the improving in activity of antioxidant enzyme could relatively reduce the content of ROS, it could not reverse the trend of high ROS content in root cells under the toxicity stress of high Mn, resulting in irreversible cell damage. In conclusion, the toxicity stress of high Mn affected the cell integrity, nutrient uptake, hormone regulation, and resistance of soybean root growth. It resulted in inhibited root growth and development and then led to the decrease in root biological yield. The regulatory process of root suffering Mn poisoning in soybean was shown in Figure 11. The results demonstrated that soybean roots may have complicated regulating mechanisms in responding to Mn poisoning stress, thus demanding to study further.” -> To me this is clearly discussion section, so I suggest move it to there.

 My response: L657-681. That's pretty good advice. I have modified it as suggested and moved it to discussion section.

13.Discussion:

General comment: Please mention the relevant figures whenever you are describing your results. It makes easier to the reader to observe what you are discussion.

 My response: L417, 418, 427, 432, 442, 450, 457, 461, 466, 483, 491, 527, 528, 540, 541, 558, 559, 573, 585, 595, 605, 613, 629, 642, 653. That's pretty good advice. I have mentioned the relevant figures and tables in the discussion section.

14.“Excessive effective Mn”. Please clarify.

 My response: L410. “Excessive effective Mn” means Mn excess or Mn poisoning.

I have changed “Excessive effective Mn” into “Mn poisoning”.

15.“APX, POD, and SOD are the primary antioxidant substances”. APX, POD and SOD are not antioxidant substances, they are antioxidant enzymes!

 My response: L476-477. That's correct. The “primary antioxidant substances” has been changed into “antioxidant enzymes”.

16.Material and Methods

4.1 Section. Was the EC of the nutrient solution controlled?

 My response: Yes. Because the culture solution was updated every 5 days. And the pH value of culture solution was regulated to 5.0 by 1 M potassium hydroxide or sulfuric acid every 2 days. Each replacement of the nutrient solution contained the same nutrient composition and concentration as those used before. So that the EC of the nutrient solution was controlled.

17.4.2 Section: “and thoroughly dried at the temperature of 105 °C for maintaining 30 min”. Are the authors sure that 30 min were enough to have the dry weight?

 My response: L718. That's good question. I have omitted to describe a drying process earlier.

“and thoroughly dried at the temperature of 105 °C for maintaining 30 min”. Therewith the samples were then kept in an oven at 65 °C for 8 days.

18.“Each experiment was repeated four times” Please specify what do you mean with this sentence. Did you perform 4 experiments? Or did you have 4 biological replicates?? Please clarify it.

 My response: L719-720. That's good question. I have added a detailed clarify.

“Each experiment was repeated four times” has been changed into “There were four biological replicates for each treatment.”

19.“The CAT activity was determined using the reported technique [108]. The mixed sub- stance contained 2.9 mL of 30% hydrogen peroxide, 0.1 mL enzyme extracted material, and 0.15 M PBS (pH=7.0). The activity of CAT was evaluated by supervisory controlling the attenuation level of the absorbance value of hydrogen peroxide in OD470 nm.” Why did the authors measured Cat activity at 470 nm if the usual absorvance is 240 nm, including in the reference that the authors mention. Please clarify it.  Please also clarify if kinetics was used (and how – time range) to quantify CAT and APX activity.

 My response: L791. Thank you for helping us correct this mistake. We have checked the previous test methods and found that the absorbance used to measure CAT activity was indeed 240 nm. This was a mistake we had made when writing papers.

20.Please briefly described the extraction method for hormone quantification.

My response: L828-855. I have added this part, as shown below.

The produced standard curve working solution may be utilized to construct the standard curve for further hormone content computation. First, 988 μL of methanol solution (Kermel, China) was placed into 1.5 mL centrifuge tubes, and 2 L of each 500 g/mL of the corresponding hormone external standard stock solution was added, shaken well, and configured as the external standard master solution with an eventual concentration of 1 μg/mL. Then, 990 μL methyl alcohol solution was supplemented in 1.5 mL centrifugal tube, and 2 L of each 500 μg/mL of the appropriate hormone internal standard stock solution was added and agitated vigorously to make the master batch of internal standard at a final concentration of 1 μg/mL. Finally, methanolic solutions were used to make standard curves with final concentrations of 0.1, 0.2, 0.5, 2, 5, 20, 50, and 200 ng/mL, with each con-centration point containing 20 ng/mL of the appropriate hormone internal standard. On the basis of the results of HPLC–MS/MS measurements, the standard curve can be plotted, where the horizontal coordinate X is the concentration of the external standard divided by the corresponding internal standard, and the vertical coordinate Y is the peak area of the external standard divided by the corresponding internal standard.

Four hormones were extracted from soybean roots by crushing samples to be analyzed in liquid nitrogen and carefully weighing 1 g of the samples in glass test tubes. Each hormone test was performed four times. A 10-fold amount of acetonitrile solution was added, as well as 8 μL of the 20 ng/mL master solution of the matching internal standard. The root samples were carefully extracted overnight at temperature of 4 °C and centrifuged for 5 min at speed of 12,000 rpm. After the liquid supernatant was obtained, the deposit was treated with a fivefold amount of acetonitrile solution, extracted two times, and combined with the supernatants. The sample received about 35 mg of C18 packing before being shocked severely for 30 seconds and centrifuged at speed of 10,000 rpm for 5 min. The liquid supernatant was then removed. The sample was blow-dried with nitrogen, redissolved in 400 μL methyl alcohol, filtered through a 0.22 μm millipore plastic membrane filter, and reserved at −20 °C in a refrigerator (Ronshen, China) for HPLC–MS/MS.

Round 2

Reviewer 1 Report

The authors have improved and incorporated all the suggestions given in my first review. I am satisfied with the qualit of this manuscript and it can be accepted now. 

Author Response

Reviewer 1

The authors have improved and incorporated all the suggestions given in my first review. I am satisfied with the quality of this manuscript and it can be accepted now.

My response: Thank you very much for your recognition and affirmation.

Reviewer 2 Report

Dear Authors,

Congratulations on your MS, which was significantly improved. There are only minor editing of English language that should be checked before publication. Please carefully read the MS and do such corrections.

Best Regards

Only minor editing of English language is now required 

Author Response

Reviewer 2

Congratulations on your MS, which was significantly improved. There are only minor editing of English language that should be checked before publication. Please carefully read the MS and do such corrections.

My response: Thank you very much for your helpful advice. I have edited the English language of the manuscript, and the changes have been marked in bright yellow or red. Please check and review.
